# Repeatability of adaptation in sunflowers reveals that genomic regions harbouring inversions also drive adaptation in species lacking an inversion

Shaghayegh Soudi[1†], Mojtaba Jahani[1,2†], Marco Todesco[2,3,4], Gregory L Owens[5], Natalia Bercovich[2], Loren H Rieseberg[2], Sam Yeaman[1]*

[1]Department of Biological Sciences, University of Calgary, Calgary, Canada; [2]Department of Botany, University of British Columbia, Vancouver, Canada; [3]Michael Smith Laboratories, University of British Columbia, Vancouver, Canada; [4]Irving K. Barber Faculty of Science, University of British Columbia Okanagan, Kelowna, Canada; [5]Department of Biology, University of Victoria, Victoria, Canada

**\*For correspondence:**
samuel.yeaman@ucalgary.ca

[†]These authors contributed equally to this work

**Competing interest:** The authors declare that no competing interests exist.

**Abstract** Local adaptation commonly involves alleles of large effect, which experience fitness advantages when in positive linkage disequilibrium (LD). Because segregating inversions suppress recombination and facilitate the maintenance of LD between locally adapted loci, they are also commonly found to be associated with adaptive divergence. However, it is unclear what fraction of an adaptive response can be attributed to inversions and alleles of large effect, and whether the loci within an inversion could still drive adaptation in the absence of its recombination-suppressing effect. Here, we use genome-wide association studies to explore patterns of local adaptation in three species of sunflower: *Helianthus annuus*, *Helianthus argophyllus*, and *Helianthus petiolaris*, which each harbour a large number of species-specific inversions. We find evidence of significant genome-wide repeatability in signatures of association to phenotypes and environments, which are particularly enriched within regions of the genome harbouring an inversion in one species. This shows that while inversions may facilitate local adaptation, at least some of the loci can still harbour mutations that make substantial contributions without the benefit of recombination suppression in species lacking a segregating inversion. While a large number of genomic regions show evidence of repeated adaptation, most of the strongest signatures of association still tend to be species-specific, indicating substantial genotypic redundancy for local adaptation in these species.

## eLife assessment

This is a **valuable** comparative study of adaptation across multiple species. The results provide a **solid** example of the application of genotype–environment associations to demonstrate that local adaptation is repeatable.

## Introduction

The genetic basis of local adaptation is sometimes highly repeatable, with examples of large effect genes driving responses in multiple species, such as *FT* affecting flowering time in numerous plants (*Izawa, 2007*; *Auge et al., 2019*) or *Mc1r* driving colour polymorphisms in vertebrates (*Manceau et al., 2010*; *Rosenblum et al., 2014*). Local adaptation can also be repeated for polygenic traits, with significant patterns of similar association found across many loci for comparisons of conifers (*Yeaman*

**eLife digest** In plants, like in humans, DNA is arranged into sections known as genes that are in turn organised into structures called chromosomes. Mutations that modify the activity of these genes can help plant species to adapt to a new environment or to extreme conditions such as drought. However, successful adaptation often requires changes in many different genes. If these sets of genes are located close to each other on the same chromosome, any mutations will likely be passed onto the next generation together. If the genes are located further away, or even on different chromosomes, they may instead be inherited separately so that the next generation does not benefit as much from the adaptation.

A chromosome inversion – when a segment of chromosome breaks off and reattaches the other way around – can increase the likelihood that sets of mutations on the same chromosome will be inherited together. Many previous studies have found that chromosome inversions tend to drive the ability of species to adapt to different environments by keeping together mutations that affect the same characteristics. However, it is not clear how inversions affect the repeatability of the adaptation, that is, if another group of closely related plants faced the same challenge in their environment would they evolve in the same way, or would they evolve a new response?

To address this question, Soudi, Jahani et al. used a genetics approach known as a genome wide association study to explore how three closely related species of sunflower have adapted to their respective environments. Two of the species grow in various environments across the centre and west of the USA that are often hot and dry, whereas the third species is restricted to the more humid coastal plain of Texas, USA.

The experiments found that a few key genes had changed in all three sunflower species. However, each species also had mutations in a larger set of unique genes that were not changed in the other species. Regions of chromosomes harbouring inversions in one of the species tended to have more of the key genes within them, compared to other genomic regions. This was also true for species that did not have inversions in those regions. This demonstrates that genes in regions affected by chromosome inversions can still help plants adapt to changes in the environment even in the absence of inversions.

Sunflowers are widely grown for their edible oily seeds. In the future, some of the key genes identified in this work may be useful candidates for plant breeding to improve the resilience of sunflowers to drought, high temperatures and other environmental challenges.

et al., 2016), maize and its wild relative teosinte (*Tittes et al., 2021*; *Wang et al., 2021*), and *Brassicaceae* (*Bohutínská et al., 2021*), to name a few. While we have a growing number of examples of repeatability in the basis of adaptation, it is also interesting to know if species use different genes to adapt to the same selection pressure. Genotypic redundancy – the potential for many genotypes to yield a given phenotype – is one critical factor affecting the repeatability of adaptation, as high redundancy would be expected to result in lower repeatability (*Yeaman et al., 2018*). Another critical factor affecting repeatability is shared standing variation, whether present due to introgression or incomplete lineage sorting. In either case, variants shared among lineages are much more likely to contribute to a repeated response than new mutations (MacPherson and Nuismer 2018; Ralph and Coop 2015). Consistent with this, *Bohutínská et al., 2021* found that repeatability was negatively related to phylogenetic distance. While we are now accumulating more studies about the repeatability of adaptation, we still have very few examples and much remains unknown about the relative importance of these factors (*Yeaman, 2022*).

Inversions have been implicated in local adaptation in many species (*Wellenreuther and Bernatchez, 2018*), likely due to their effect to suppress recombination among inverted and non-inverted haplotypes, and thereby maintain linkage disequilibrium (LD) among beneficial combinations of locally adapted alleles (*Rieseberg, 2001*; Noor et al. 2001; *Kirkpatrick and Barton, 2006*). This has been approached by models studying the establishment of inversions that capture combinations of locally adapted alleles present as standing variation (e.g. *Kirkpatrick and Barton, 2006*), as well as models examining the accumulation of locally adapted mutations within inversions (e.g. *Schaal et al., 2022*). If there is variation in the density of loci that can potentially contribute to local adaptation, inversions

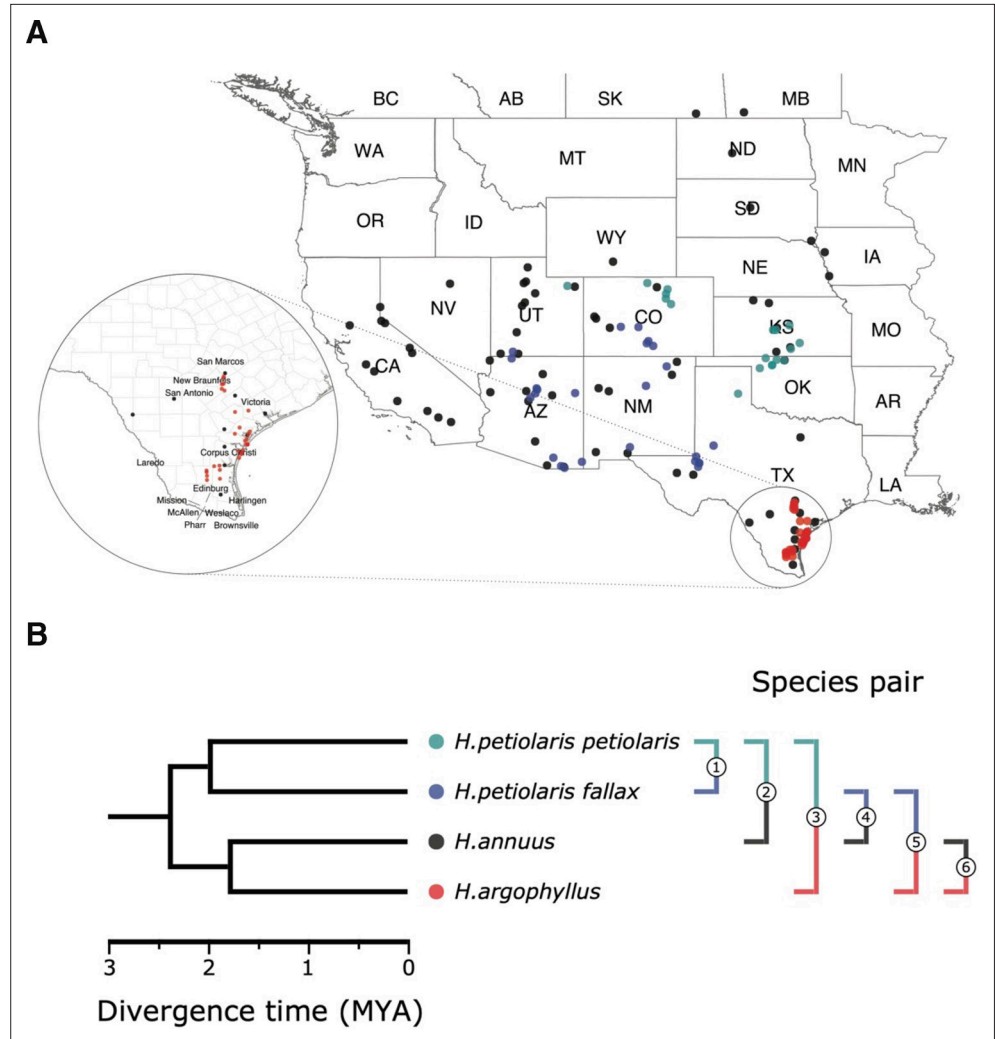

**Figure 1.** Sampling sites and phylogenetic relationship among surveyed species. (**A**) Sampling locations of wild sunflower populations studied in this study, and (**B**) phylogenetic relationship of the four (sub-)species. Numbered brackets represent the six pairwise comparisons performed in this study.

would be expected to preferentially establish and be retained in regions harbouring a high density of such loci (and this expectation would hold for both the capture and accumulation models). We would also expect to see stronger signatures of repeated local adaptation in such high-density regions. Despite mounting evidence of their importance in adaptation, it is unclear how inversions may covary with repeatability of adaptation among species. A fundamental parameter of importance in these models is the relationship between migration rate and strength of selection on individual alleles, which may not make persistent contributions to local adaptation without the suppressing effects of recombination if selection is too weak (*Yeaman and Whitlock, 2011*; *Bürger and Akerman, 2011*). If most alleles have small effects relative to migration rate and can only contribute to local adaptation via the benefit of the recombination-suppressing effect of an inversion, then we would expect little repeatability at the site of an inversion – other species lacking the inversion would not tend to use that same region for adaptation because selection would be too weak for alleles to persist. On the other hand, if some loci are particularly important for local adaptation and regularly yield mutations of large effect, with these patterns being conserved among species, repeatability within regions harbouring inversions may be substantial. Thus, studying whether adaptation at the same genomic region harbouring an inversion is observed in other species lacking the inversion can give insights about the underlying architecture of adaptation, and the evolution and maintenance of inversions.

Here, we explore the repeatability of local adaptation in three species of sunflowers, *Helianthus annuus*, *Helianthus argophyllus,* and *Helianthus petiolaris* (***Figure 1***), which harbour large regions of suppressed recombination ('haploblocks'), most of which are inversions, and are often associated with adaptive traits (***Todesco et al., 2020***). *H. annuus*, the common sunflower, is the closest wild relative of cultivated sunflower, which was domesticated from it around 4000 years ago (***Blackman et al., 2011***). Populations of *H. annuus* are distributed throughout the central and western United States and generally found on mesic soils, but can grow in a variety of disturbed or extreme habitats, such as semi-desertic or frequently flooded areas, as well as salt marshes. *H. petiolaris*, the prairie sunflower, prefers sandier soils, and ecotypes of this species are adapted to sand sheets and sand dunes (***Ostevik et al., 2016***). Here we include samples from two subspecies: *H. petiolaris* ssp. *petiolaris*, which is commonly found in the southern Great Plains, and *H. petiolaris* ssp. *fallax*, which is limited to more arid regions in Colorado, Utah, New Mexico, and Arizona (***Heiser et al., 1969***). *H. petiolaris* and *H. annuus* have broad and overlapping distributions throughout the central and western United States and appear to have adapted to similar changes in temperature, moisture, and photoperiod regimes. There is also evidence indicating *H. annuus* and *H. petiolaris* have likely been exchanging genes during much of their history of divergence (***Strasburg and Rieseberg, 2008***), although partially isolated by strong pre- and post-zygotic barriers (***Sambatti et al., 2012***). The third species, *H. argophyllus*, the silverleaf sunflower, is found exclusively in the southeast coast of Texas and includes both an early flowering ecotype on the coastal barrier islands and a late flowering ecotype inland (***Moyers and Rieseberg, 2016***; ***Todesco et al., 2020***). *H. argophyllus* is thought to have undergone cycles of sympatry and allopatry with *H. annuus* and *H. petiolaris* over time, but currently only overlaps with *H. annuus,* with thus overlap likely being a recent event (***Heiser, 1951***).

Using broad sampling across the ranges of these species (***Figure 1***) and sequencing data first published by ***Todesco et al., 2020***, we study the genomic basis of local adaptation by conducting genome-wide scans for associations with climatic and soil environmental variables and phenotypes measured in a common garden. An inherent problem in studying the basis of local adaptation is accounting for the covariance between genome-wide population genetic structure and environment. When the selection pressure driving local adaptation tends to parallel a main axis of demographic expansion or isolation by distance, neutral alleles will have spatial patterns resembling those of causal alleles. Methods that do not correct for population structure have large numbers of false positives because allele frequencies at neutral loci tend to correlate with the environment more than expected by chance (***Lotterhos and Whitlock, 2014***). By contrast, methods that use structure correction tend to suffer from false negatives because the causal loci have similar patterns of allele frequency variation as the genomic background, and so their statistical signatures are decreased (***DeRaad et al., 2021***; ***Booker et al., 2023a***). When a non-corrected approach is applied to multiple species, the false-positive problem can be mitigated when testing for loci that contribute to repeated adaptation as the same gene should not tend to be a false positive in multiple species more often than expected by chance (***Yeaman et al., 2016***). Here we use a conventional genome-wide association study (GWAS) approach with structure correction to study the basis of phenotypes, which vary both within and among populations, and use an uncorrected approach to test association to environment, which only varies among populations (as all individuals within the same population have the same value of environmental variable). For the environmental association portions of the study, we also explore the effect of structure correction in some analyses to contrast the false-negative vs. false-positive problem inherent in each approach. We apply these approaches individually to *H. annuus*, *H. argophyllus*, and the two *H. petiolaris* subspecies, and make pairwise and higher-order comparisons among combinations of these four lineages to study repeatability.

Our overall aim is to characterize the extent of repeatability of local adaptation for different traits and environments and explore the importance of inversions and other factors that may drive differences in repeatability. This analysis builds upon the work of ***Todesco et al., 2020***, which focused on the evolution of haploblocks within each species, but did not systematically study patterns of repeatability in signatures of association among species. We begin by using the strength of correlations between environment and phenotypes measured in a common garden to identify which environmental variables are likely driving local adaptation within each species, and use an index to quantify the relative similarity in these patterns among pairs of species (see ***Appendix 1—figure 1*** for a schematic overview of methods and research questions). We then conduct genome-wide association

tests to identify regions most strongly associated with phenotypes and environments within each species. To assess the similarity in these association statistics among species, we use methods similar to *Yeaman et al., 2016* to identify regions of the genome that exhibit greater similarity in signatures of association among pairs of lineages than expected by chance, which are likely driven by repeated adaptation using the same genes (but not necessarily the same alleles). We then explore how the number and total size of these regions of repeated association covary with the index of similarity in environmental importance. We also explore whether these regions of repeated association tend to covary with patterns of shared standing variation and test if they are enriched within previously identified haploblocks to explore the role of inversions in adaptation. We then identify candidate genes that show particularly strong signatures of repeated association across multiple lineages. Finally, based on the number of signatures of association that are shared vs. non-shared among lineages, we estimate the number of regions genome-wide that could potentially contribute to local adaptation for each variable and phenotype, which is related to the genotypic redundancy. Taken together, our

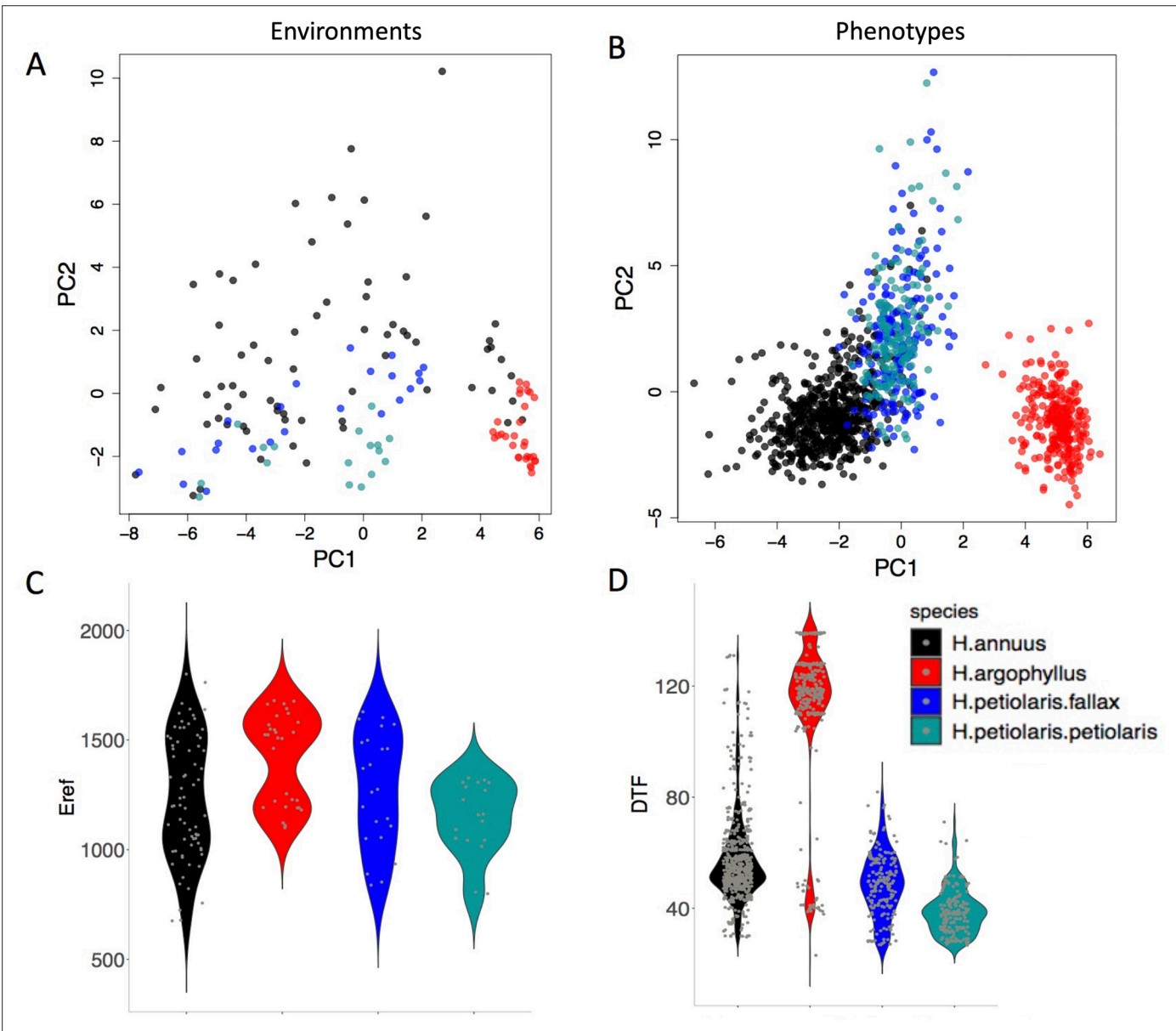

**Figure 2.** The range of environmental and phenotypic variation in the studied species. Variation in environment (**A**) and phenotype (**B**) for the studied species, along the two largest axes of a principal component analysis (PCA). Violin plots show two examples of variation in environment (Hargreaves reference evapotranspiration index; Eref) and phenotype (Days to Flower; DTF) within and among the taxa (**C, D**).

results show that local adaptation in sunflowers tends to involve both strong repeatability at a small number of genes, often associated with inversions, coupled with high redundancy and non-repeated responses across a much larger number of loci.

## Results

### Local adaptation at the phenotypic level

The four taxa vary considerably in the breadth of environmental variation spanned by their respective ranges, with *H. annuus* spanning the widest niche and *H. argophyllus* spanning the narrowest (*Figure 2A*). Using the correlation between phenotype and environment as a metric of the strength of local adaptation on individual traits, we observe strong correlations for many combinations (*Appendix 1—figure 2*), suggesting that local adaptation is quite pronounced for some traits. To quantify the similarity among pairs of species in the strength of local adaptation at the phenotypic level (and to later compare this to results at the genomic level), we calculate an index we refer to as the similarity in phenotype–environment correlation (SIPEC), which is maximized when both species have strong correlations (in either direction) between a phenotype and environmental variable (see 'Materials and methods'). For each environmental variable, we take the maximum value of SIPEC across all phenotypes as a relative measure of the importance of the variable for driving local adaptation in both species (such that when SIPEC is low, the variable is not a strong driver of phenotypic adaptation in at least one of the two species). We find the largest values of SIPEC for temperature variables and the smallest values for soil types, and generally find higher values for comparisons between the *petiolaris* subspecies (*Appendix 1—figure 3*).

### Genome-wide analysis of repeated local adaptation

To search for regions of the genome driving repeated adaptation in multiple taxa, we first identified windows of the genome within each species that showed strong signatures of association to either phenotype (GWAS; with structure correction) or environment (GEA; without structure correction), referred to as 'top candidate' windows. To identify windows of repeated association (WRAs) between pairs of species, we then assessed whether top candidate windows identified in one focal taxon also tended to be enriched for strong signatures of association to the same environment or phenotype in each of the other taxa (using the null-W test, which tends to be more sensitive than just finding the overlap between the top candidate windows; see 'Materials and methods'). As recombination rate can affect the sensitivity of these kinds of window-based genome scans (*Booker et al., 2020*), the identification of WRAs was conducted after binning windows by recombination rate, although in many cases we did not observe substantial differences in the null distributions among these bins (e.g. *Appendix 1—figure 4*). This analysis revealed many windows with signatures of repeated association, with the strongest repeated signature found for number of frost-free days (NFFD; *Figure 3A*; *Appendix 1—figure 4*), but also showed that many of the windows with the strongest association in one species did not have strong signatures in other species (*Figure 3B*).

Under the null hypothesis that all regions of the genome evolve neutrally due to drift and independently in each species, ~5% of the top candidate windows identified in one species would be expected to fall into the tail of the null distribution of another species (i.e. be classified as WRAs). While we find significantly more WRAs than the 5% expected by chance, ranging from 6.3 to 44.1% (mean = 14.5%; *Appendix 1—figure 5*), multiple factors can violate the assumptions of this test and increase the proportion of windows classified as WRAs. Most importantly, similarity in signatures of association can be driven by shared ancestral variation or ongoing introgression, and this must always be considered as an alternative explanation. It is clear that most regions of the genome harbour some shared standing variation between all six pairwise comparisons of the taxa, whether due to segregating ancestral variation or introgression, based on an index quantifying the proportion of shared to non-shared SNPs in each window (*Appendix 1—figure 6*). However, we do not find consistent differences in the amount of shared standing variation within windows when comparing WRAs with the rest of the genome (*Appendix 1—figure 6*). Thus, while introgression may make subtle contributions to WRAs, a lack of increased shared standing variation in the significant windows suggests that it is not a primary driver of these patterns.

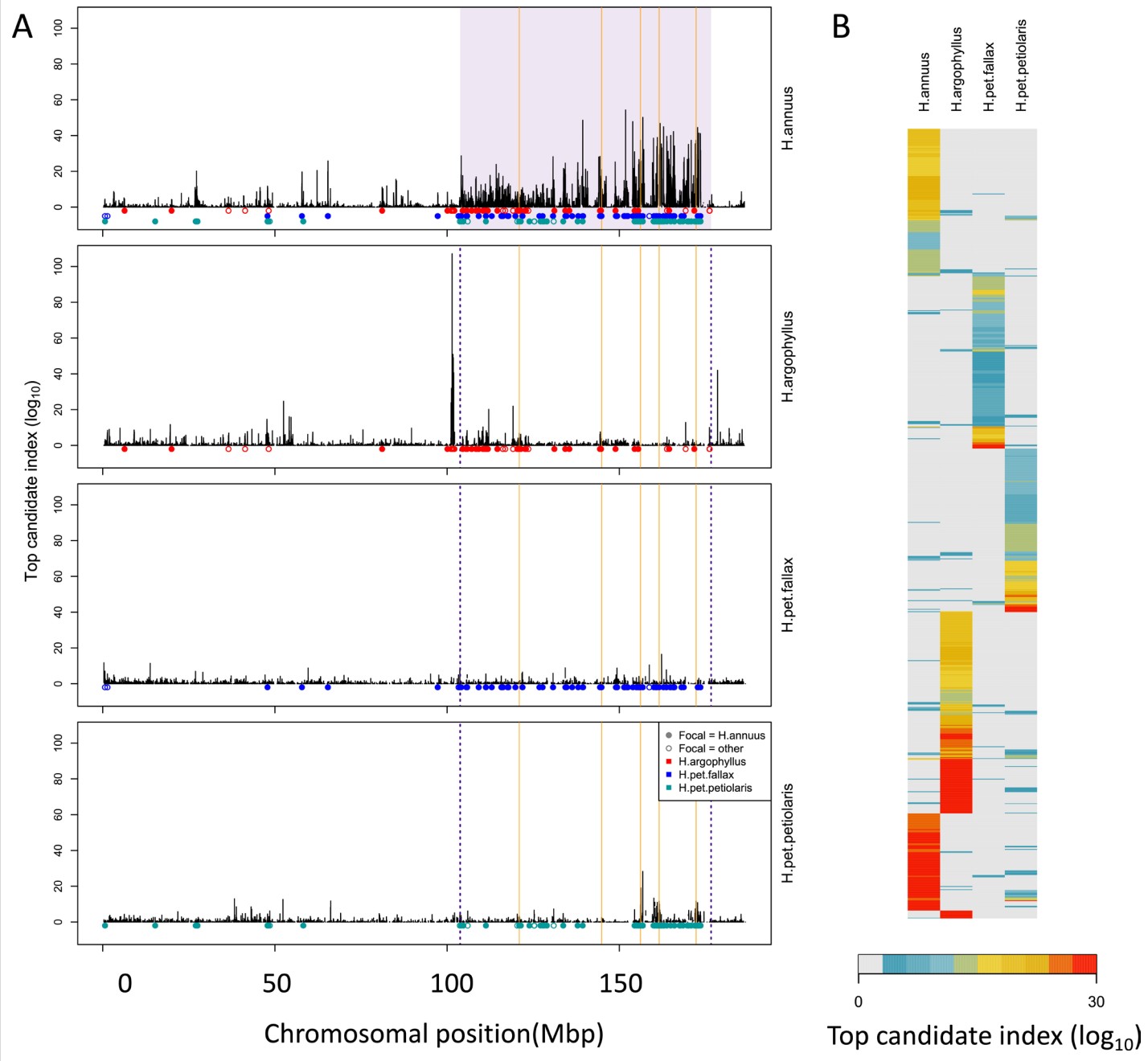

**Figure 3.** Signatures of association for number of frost-free days (NFFD) in the four taxa on chromosome 15 (**A**) and genome-wide (**B**). Panel (**A**) shows windows of repeated association (WRAs; coloured bars) for comparisons between the focal species, *H. annuus*, and each of the other three taxa, with the haploblocks in *H. annuus* shaded in violet and the regions with significant PicMin hits as vertical orange lines. Panel (**B**) shows the value of the top candidate index for each of the 1000 windows with the strongest signatures of association in at least one species (approximately top 2% of genome-wide windows). Rows are ordered using hierarchical clustering to group windows with similar patterns across multiple species, illustrating the extent of overlap/non-overlap in the windows with strongest signatures in each species (i.e. position in the figure does not reflect chromosomal position).

A second source of inflation in the number of WRAs is due to hitchhiking: if a given locus is a true positive driving adaptation, LD with other windows in tight physical linkage will result in spurious correlations causing them to also be classified as WRAs increasing the genome-wide proportion above 5%, and this effect will be particularly exaggerated in haploblocks. To control for this, we binned neighbouring WRAs together that exhibited high LD (>95th percentile) over short spans of the genome (<1 cM) in either species, yielding regions we refer to as clusters of repeated association

(CRAs). While this clustering method should provide a partial control for the effect of hitchhiking, we note that the number of CRAs cannot be taken as a reliable estimate of the number of independent targets of natural selection. Instead, we treat the number and size of these CRAs as proxies for the relative genome-wide similarity in association between each pair of species, and conduct downstream analyses to test hypotheses about the factors driving local adaptation.

CRAs varied in number and size across the six pairwise-species comparisons, with particularly large CRAs identified at the sites of known haploblocks (*Todesco et al., 2020*). For CRAs identified for phenotypic associations, their extent ranged from 168 clusters spanning 48,524,903 bp (covering only 1.6% of the genome) in *H. argophyllus-H. pet. petiolaris* to a maximum of 1667 clusters spanning 426,409,647 bp (14.2% of the genome) in *H. pet. petiolaris-H. pet. fallax* (*Appendix 1—figure 7*). CRAs identified for environmental variables tended to be more numerous and cover a larger region of the genome, varying from 1154 clusters comprising 15% of the total genome for *H. argophyllus-H. pet. fallax* up to 2260 distinct clusters covering 29% of the genome between *H. petiolaris* subspecies (*Appendix 1—figure 8*). The large extent of the genome covered by CRAs is mainly driven by their occurrence within many different haploblocks, which tend to be unique to each species and cover substantial portions of their genomes (*Todesco et al., 2020*). Genes within CRAs tend to be enriched for a large number of GO terms spanning many different categories (*Appendix 1—figures 9 and 10*). When we use an approach that corrects for population structure in the genotype–environment association tests (BayPass), we find substantially reduced numbers of CRAs (*Appendix 1—figure 11*). This reduction is likely due to reduced power to detect true positive causal loci when population structure covaries with environment, as true positives do not 'stand out' relative to the genomic background (*Booker et al., 2023a*). While single-species association tests without correction for population structure typically yield large numbers of false positives, the null-W test comparing association results across species accounts for the chance that the same region is a false positive in both species (*Yeaman et al., 2016*).

## Repeated association in the genome reflects patterns at the phenotypic level

If natural selection is driving repeated patterns of local adaptation in the genomes of two species, we should see greater similarity for associations with environmental variables that are also strongly associated with phenotypic variation in both species. Consistent with this prediction, we found that environmental variables with a high maximum SIPEC index (i.e. high correlation between environment and some axis of phenotypic variation in both species; *Appendix 1—figure 3*) also tended to have a larger number of CRAs (*Figure 4*), with higher repeatability at both the phenotype and genome level for temperature-related variables and much lower repeatability for soil-related variables (with similar patterns for mean SIPEC index; *Appendix 1—figure 12*). Unfortunately, given the non-independence of environmental variables there is pseudo-replication in these data so it is not possible to conduct a formal significance test of these patterns. In all cases, the number of CRAs exhibited a weak negative relationship with the index of standing variation, with environmental variables that had the greatest number of CRAs having the lowest relative amounts of shared standing variation (*Appendix 1—figure 13*). This suggests that the observed similarity in patterns of association among species is not being strongly driven by incomplete lineage sorting or introgression, as we would expect a positive relationship between the index of shared standing variation and the number of CRAs. We found similar but weaker patterns in the association between SIPEC and size of CRAs (*Appendix 1—figure 14*). The weakening of these patterns relative to those for number of CRAs may reflect comparatively greater contribution of haploblocks driving patterns with size rather than number (i.e. if a particularly large haploblock is a CRA, then that variable will have a particularly large value relative to its actual importance for local adaptation).

## Overlap of signatures of repeated association with low recombination haploblocks

Inversions can facilitate local adaptation by suppressing recombination between locally adapted alleles within the inverted vs. ancestral haplotypes (*Kirkpatrick and Barton, 2006*). If a given species exhibits a signature of association to environment at a segregating inversion, it is interesting to test whether similar signatures of association are found in other species that lack a segregating inversion at

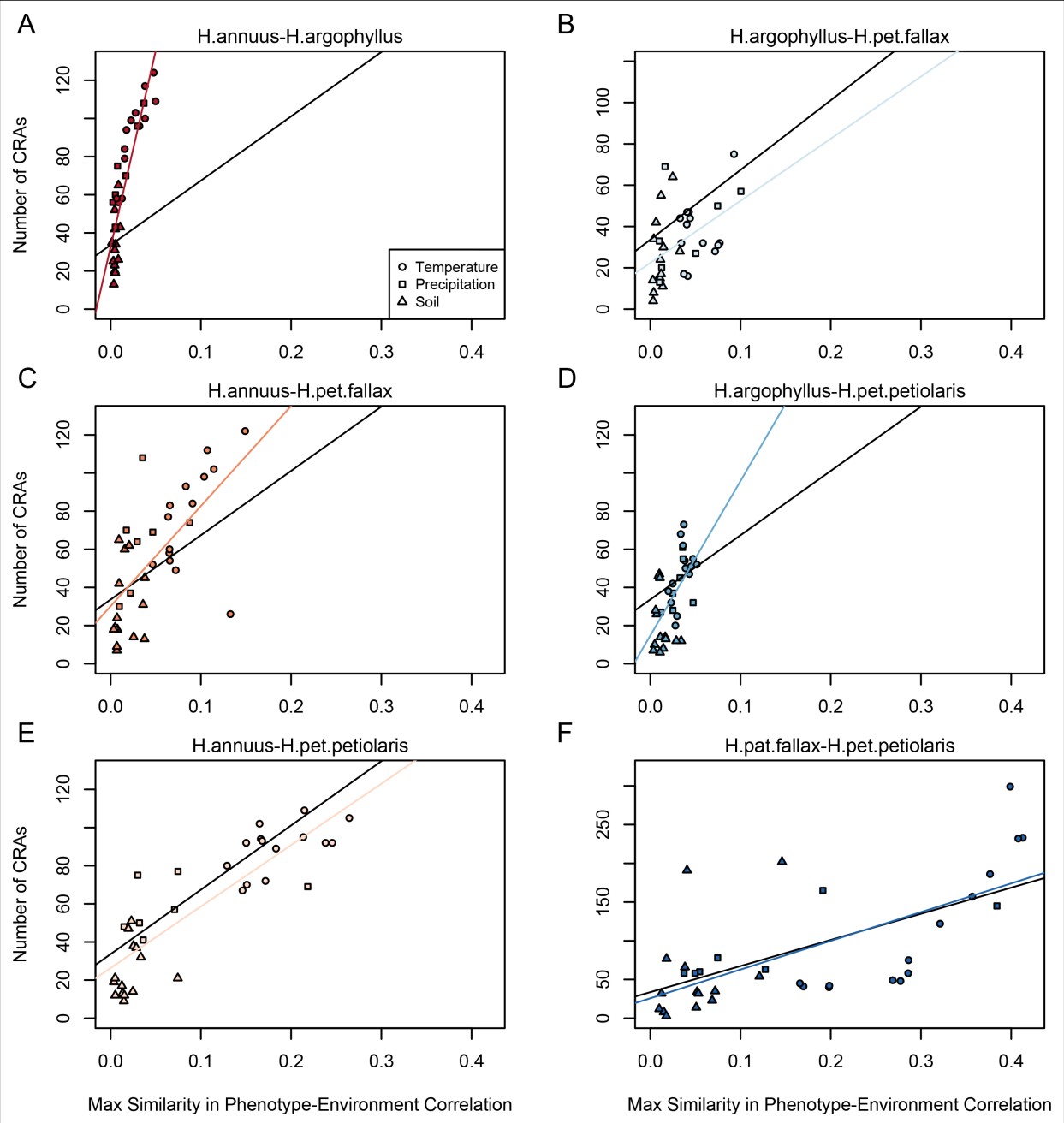

**Figure 4.** Relationship between maximum similarity in phenotype–environment correlation (SIPEC) and number of clusters of repeated association (CRAs). SIPEC was calculated for each phenotypic principal component analysis (PCA) axis, with the maximum taken across the axes that cumulatively explain 95% of the phenotypic variance for each environment. Each panel (**A-F**) shows a comparison between a pair of species indicated above, and includes both a linear model fit to the data within the panel (coloured lines), and a linear model fit to all data simultaneously (black lines) for comparison. Note that because environmental variables are correlated, these points are not independent and therefore represent a source of pseudoreplication, preventing formal statistical tests of this relationship.

the same region of the genome. This would suggest that particularly strong selection is acting on loci in this region, as signatures of association can still evolve even without the recombination suppressing effect of the inversion. While not all of the low-recombination haploblocks identified by *Todesco et al., 2020* have been validated as inversions, for simplicity we treat each haploblock as representative of a segregating inversion. The majority of these haploblocks are present as segregating variation in only one of the three species, but when they occur in *H. petiolaris*, they tend to be found as segregating variation within both subspecies. Thus, if we find significant enrichment of the regions of

repeated association (WRAs and CRAs) within haploblocks, this suggests that both the species with a segregating inversion and the species lacking an inversion are using this region of the genome to drive local adaptation. As a first test of this relationship, we assessed two-way contingency tables for whether top candidate windows within a species were also significant WRA (WRA/non-WRA) and whether they fell within haploblocks (yes/no). This approach controls for the potential enrichment of

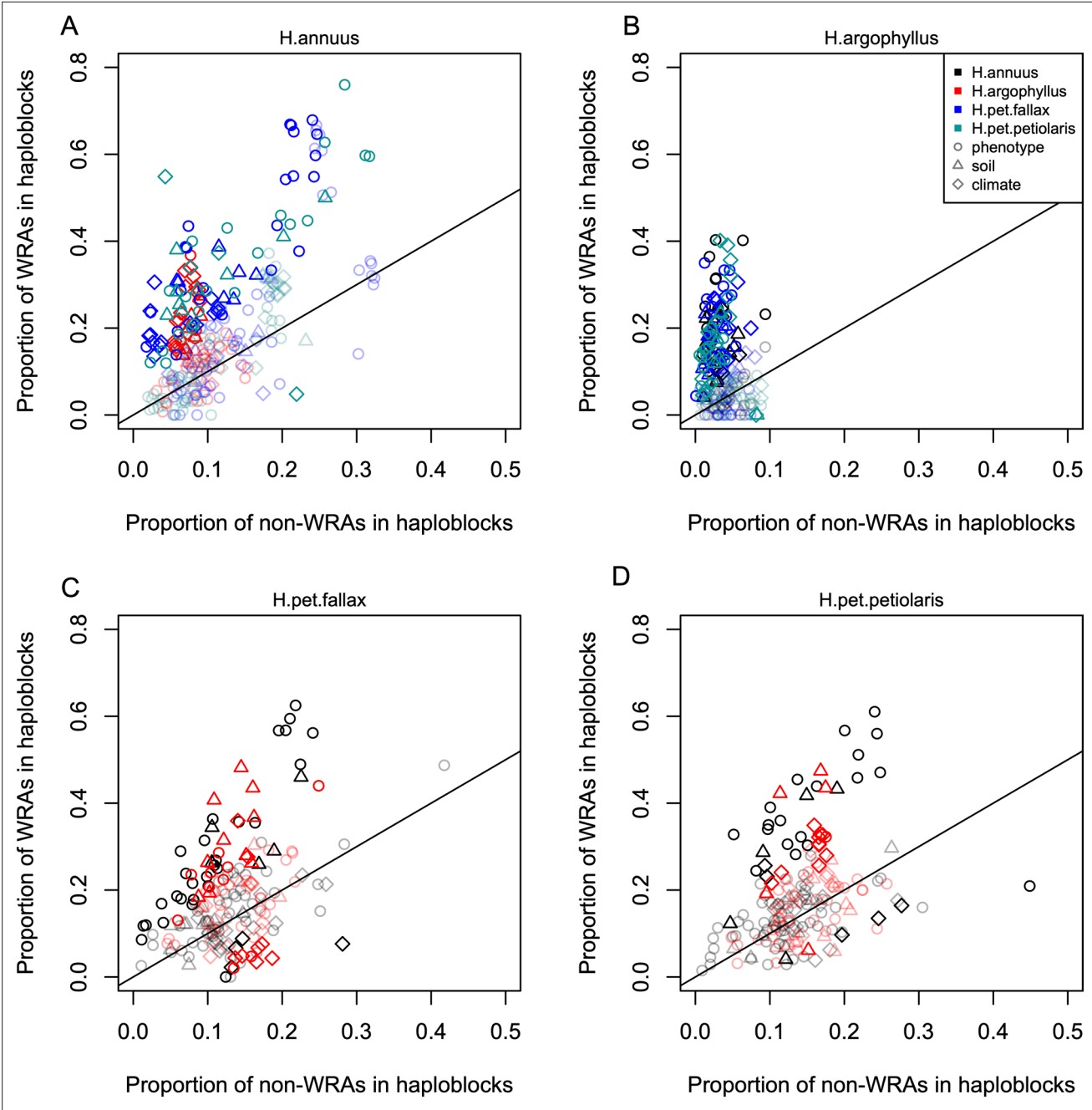

**Figure 5.** Enrichment of signatures of repeated association within genomic regions harbouring a haploblock in one of the two compared lineages. Each panel shows the proportion of top candidate windows that fall within haploblocks for windows with significant signatures of repeated association by the null-W test (windows of repeated association [WRAs]) vs. those with non-significant signatures (non-WRAs), with a different focal species plotted in each panel (**A-D**). Comparisons of *H. petiolaris petiolaris* vs. *H. petiolaris fallax* are omitted as they share segregating haploblocks. Each point corresponds to the results for a single phenotype or environment, with dark shading used for cases where the deviation from random for the contingency table is significant by a permutation test ($p<0.05$), and lighter shading indicating a non-significant result. Note that because many environmental variables and phenotypes are correlated with each other, these points are not independent and therefore represent a source of pseudoreplication, preventing formal statistical tests of the overall relationship within each panel.

top candidate windows within haploblocks relative to the rest of the genome and asks if the proportion of WRAs within haploblocks is even higher than non-WRA top candidate windows. For most phenotypes and environments, we found that the proportion of WRAs occurring within haploblocks was much higher than the proportion of non-WRAs occurring within haploblocks (*Figure 5*). Across all species comparisons and environmental/phenotypic variables, a total of 310 contrasts showed significant enrichment for WRAs in haploblocks compared to only 23 contrasts showing a significantly higher proportion of non-WRAs in haploblocks (note that because of non-independence of the phenotypes and environmental variables, there is pseudo-replication inherent in these estimates). In *H. argophyllus*, while non-WRA top candidates tended to fall within haploblocks less commonly than in the other taxa, WRAs tended to be very strongly enriched within haploblocks (*Figure 5B*). As a follow-up, we also tested whether CRAs were enriched within haploblocks (without controlling for whether top candidate windows also tended to be enriched), also finding strongly significant enrichment of CRAs within haploblocks for many traits and environments (*Appendix 1—figure 7C and D*). Broad patterns revealed by these analyses are similar as both show that genomic regions harbouring haploblocks tend to be enriched for signatures of association to environment in species lacking the haploblock. It is noteworthy that few significant signatures of enrichment in haploblocks were found for *H. petiolaris petiolaris* vs. *H. petiolaris fallax* (*Appendix 1—figure 7C and D*), perhaps because extensive introgression across non-locally adapted regions of the genome obscures true signal.

## Multi-species signatures of repeated association

As a complement to the pairwise analyses described above, we also conducted an analysis using PicMin, which simultaneously considers any number of lineages to identify genes with particularly strong and repeated signatures of association (*Booker et al., 2023b*). After identifying significant windows clustering together those <1 Mbp apart, we found a total of 145 regions that were significant for at least one environmental variable or phenotype (at False Discovery Rate (FDR) < 0.1 applied within each variable). The largest number of significant regions for environment was found with the NFFD variable (20 with FDR < 0.1; 44 with FDR < 0.2) and for phenotypes with total leaf number, a measure of developmental timing of flowering (9 with FDR < 0.1; 19 with FDR < 0.2). Considering the 720 individual 5000 bp windows with FDR < 0.2 for at least one variable, 387 (53.8%) are within 500 bp of a genic region, representing a 2.3× enrichment relative to the rate for windows that were not significant hits using PicMin (23.3% of other windows fall within 500 bp of a genic region; $X^2$ test p<10$^{-15}$).

## Estimating the number of potentially adaptive loci

Even for the environment with the highest repeatability (NFFD), most of the strongest signals of association are found in only one lineage (*Figure 3B*). Unfortunately, while the null-W test provides strong support and controls false positives when identifying repeated association, regions of the genome with strong associations in only a single species may include a large (and unknown) number of false positives. Thus, while it is biologically interesting to know how much adaptation is non-repeated among species, it is difficult to quantify with certainty (*Booker et al., 2023b*; *Booker et al., 2023a*). Under the assumption that at least some of the non-repeated signatures of association are true positives, this implies that there is considerable genotypic redundancy, with many different ways for these species to adaptively respond to variation in the same environment. To estimate the number of windows that could potentially contribute to local adaptation for each variable ($L_{eff}$), we modified the method of *Yeaman et al., 2018* to partially account for the effect of linkage among nearby windows (see 'Materials and methods'). This method assumes that the windows with signatures of association in each species are a random draw from a larger number of potentially contributing windows ($L_{eff}$), which can be inferred based on the ratio of shared vs. non-shared windows with strong association signatures (see 'Materials and methods'). We find that estimates of $L_{eff}$ tend to be large, always well over 1000 windows regardless of the trait or environment (*Appendix 1—figure 15A*). There are numerous sources of error that affect the estimation of $L_{eff}$, which will tend to be overestimated due to linkage (*Appendix 1—figure 15A*) or when many signatures of association are false positives (see Appendix 1), and will be underestimated when some repeatability is due to shared standing variation (*Yeaman et al., 2018*). Even after controls for linkage and assuming a high false-positive rate of 80%, estimates

of $L_{eff}$ remain in the hundreds even for the variable with the lowest value of $L_{eff}$ (Eref; *Appendix 1—figure 15C and D*).

## Discussion

Despite its critical importance in shaping the architecture of adaptation, little is known about the extent of genotypic redundancy underlying different traits (*Barghi et al., 2020*; *Láruson et al., 2020*; *Yeaman, 2022*). Here, we have shown evidence of significant repeatability in the basis of local adaptation (*Figures 4 and 5*), but also an abundance of species-specific, non-repeated signatures (*Figure 3*; *Appendix 1—figure 14*). In particular, we find that regions of the genome that harbour inversions in one species also tend to be strongly enriched for signatures of association in other species lacking the inversion (*Figure 5*). Taken together, this suggests that local adaptation in these species is highly flexible – different species apparently use quite different sets of loci to adapt to the same environment – yet still involves some component that has minimal redundancy, with inversions playing a particularly important role. Some of the 'usual suspects' show up in the set of significantly repeated loci identified by PicMin: in addition to the homologs of the *FT* (*FLOWERING LOCUS T*) gene reported by *Todesco et al., 2020*, we also found hits for several other genes involved in circadian regulation and flowering time, including *PRR3* (*Para et al., 2007*), *TOE1* (*Aukerman and Sakai, 2003*), and *PHYC* (*Takano et al., 2005*; *Chen et al., 2014*), which is known to regulate photoperiodic responses in *Arabidopsis* accessions (*Balasubramanian et al., 2006*). Other top hits included genes involved in plant development and auxin transport (*PIN3* [*Keuskamp et al., 2010*], *ARF4* [*Pekker et al., 2005*], and *MN* [*Bhatia et al., 2016*]), and plant immunity (*CRK13* [*Acharya et al., 2007*] and *PRR2* [*Cheval et al., 2017*]).

The increased repeatability found in regions of the genome that harbour inversions in only one species is particularly interesting. Inversions are commonly associated with local adaptation (*Wellenreuther and Bernatchez, 2018*), likely because they reduce the rate at which recombination breaks up combinations of co-selected alleles (*Kirkpatrick and Barton, 2006*), which perhaps facilitates contributions by alleles that would be individually too weakly selected to overcome swamping by migration (*Bürger and Akerman, 2011*; *Yeaman and Whitlock, 2011*; *Schaal et al., 2022*). Evidence here suggests these regions still tend to contribute to adaptation in the species lacking the recombination-suppressing effect of an inversion, consistent with a strong effect of selection relative to migration on at least one locus in the region (rather than adaptation exclusively via many alleles of small effect; *Yeaman, 2022*). As an example, chromosome 15 harbours a large (72 Mbp) haploblock in *H. annuus* that is strongly associated with NFFD, and also shows some signatures of association in the other taxa, with particularly strong signatures of association on either side of the haploblock in *H. argophyllus* (*Figure 3A*). Interestingly, loci of repeated association identified by PicMin within this region include two genes whose homologs are known to regulate responses to cold: *COLD-RESPONSIVE PROTEIN KINASE 1,CRPK1* (*Liu et al., 2017*) and *LATE ELONGATED HYPOCOTYL, LHY* (*Mizoguchi et al., 2002*; *Dong et al., 2011*). It seems likely that strong selection relative to migration is therefore acting upon several loci in this region, and in many others harbouring inversions.

The observed repeatability associated with inversions further supports the local adaptation model as an explanation for the long-term persistence of segregating inversions (at least in sunflower), rather than mechanisms based on dominance or meiotic drive (*Rieseberg, 2001*). If there is variation across the genome in the density of loci with the potential to be involved in local adaptation, then the establishment and maintenance of inversions would be biased towards regions harbouring a high density of such loci under this model. If the genomic basis for local adaptation is conserved amongst species, then these same regions are more likely to have high repeatability. Thus, our observation of genomic regions harbouring inversions also being enriched for WRAs is consistent with this general model for inversion evolution. Unfortunately, our observations do not provide much insight into whether inversions evolve through the capture (e.g. *Kirkpatrick and Barton, 2006*) or accumulation (e.g. *Schaal et al., 2022*) type of model as either model would be consistent with our results. Most of the sunflower inversions are >1 My old, and therefore predate any current local adaptation patterns, but likely do not predate the genes underlying local adaptation (which appear to be shared among the species we studied). As for the alleles underlying local adaptation, they may be younger than the inversions, but as our work suggests, these regions are prone to harbouring locally adaptive alleles, so it is possible that they also harboured other ancestral locally adaptive alleles. While many studies have demonstrated the importance of inversions for adaptation (*Wellenreuther and Bernatchez, 2018*;

*Hager et al., 2022*), to our knowledge only two other studies have documented the involvement of the same loci making contributions in the absence of the recombination-suppressing effect of the inversion (*Lee et al., 2017*; *Coughlan and Willis, 2019*). This also highlights how comparative studies of a species lacking an inversion may help identify which genes are driving adaptation in another species with an inversion, as segregating inversions tend to have extensive LD that prevents identification of any potential targets of natural selection within them.

While our results suggest a large number of loci can potentially contribute to adaptation, implying high redundancy (*Figure 3B*; *Appendix 1—figure 14*), there are several factors that complicate inference. Separating the effects of drift and selection to detect signatures of local adaptation is notoriously difficult because population structure often covaries with features of the environment that drive adaptation (*Lotterhos and Whitlock, 2014*; *Hoban et al., 2016*; *DeRaad et al., 2021*). As found by other analyses (*Yeaman et al., 2016*; *DeRaad et al., 2021*), when we use structure correction in our genotype–environment association tests, we find many fewer signatures of repeated association (*Appendix 1—figure 7* vs. *Appendix 1—figure 11*), likely due to the reduced power of BayPass to detect true positives when the environment covaries with population structure (*Booker et al., 2023a*). Here, we have side-stepped the issues involved in correction for population structure by instead relying on comparisons among species to identify loci with associations more extreme than expected by chance in multiple species, using the null-W and PicMin tests. This assumes that most loci in the genome are not involved in local adaptation, so that the relatively small proportion that are driving adaptation can therefore be picked out due to their tendency to fall into the tail of the distribution of association statistics in at least one other species (which would only occur at the rate of random chance under a purely neutral model). While this approach should be relatively robust for identifying loci with repeated patterns of adaptation, there is no way to formally estimate significance of associations found in only a single species, many of which may be false positives due to covariation of population structure and environment. However, even if we assume that 80% of the observed non-repeated loci are false positives, we still find that estimates of the effective number of loci contributing to adaptation ($L_{eff}$) are in the hundreds (*Appendix 1—figure 15C*). As we are unable to detect loci of small effect due to the limited power, but these loci still likely make important contributions to heritability, our estimates of $L_{eff}$ will also be biased downwards by excluding these potentially important drivers. Finally, it is also difficult to exclude the contribution of introgression or incomplete lineage sorting to the observed signatures of repeated association. If a locus tended to be highly introgressed between two species in a restricted region of their range, it is possible it could also covary with environment and therefore result in a signature of repeated association that would be mistakenly interpreted as adaptation. While it is difficult to preclude this from our analysis, regions of the genome with signatures of repeated association do not tend to have higher levels of shared standing variation than background regions (*Appendix 1—figure 6*), suggesting that this is not a broad explanation for our observations. If anything, shared standing variation would be expected to increase repeatability of adaptation (*MacPherson and Nuismer, 2017*), adding further weight to the inference that there is high genotypic redundancy in these species.

In general, we find that temperature is the strongest driver of repeated adaptation at both the phenotypic and genomic levels. We quantified local adaptation at the phenotypic level using correlations between traits measured in common gardens and the home environment of the population they were sampled from. Across all phenotypes, these correlations tended to be strongest and most similar among species for temperature variables, particularly in comparisons among *H. annuus* and *H. petiolaris* subspecies (*Appendix 1—figure 3*). We see similar patterns of repeatability reflected in the genome, where temperature variables also tend to have the greatest repeatability (*Figure 4*). The similarity in these phenotypic and genomic signatures is consistent with an effect of strong selection as other artefactual or drift-based explanations for repeatability would not be expected to reflect patterns found at the phenotypic level. It should be noted that the reduced importance of soil variables in the SIPEC index might be partly driven by the fact that all traits were measured on aboveground features due to the difficulty of getting non-disruptive phenotypic measurements for roots.

Taken together, these results suggest that some fraction of the genome contributes to adaptation with low redundancy and high repeatability (which tends to be enriched within genomic regions where there exist inversions in at least one species), while the remainder of the adaptive response is driven by loci with high redundancy and species-specific contributions. Theoretical models of adaptive

evolution necessarily involve simplified representations of genetics: population genetic approaches explore cases where strong selection deterministically drives a change in allele frequency at one or a few loci (usually without epistasis), whereas quantitative genetic approaches make the infinitesimal assumption that phenotypic change can be realized through small frequency changes at many loci (*Barton et al., 2017*; *Barghi et al., 2020*; *Yeaman, 2022*). Such models implicitly make quite different assumptions about redundancy: if a population genetic model of directional selection includes no epistasis, then it is implicitly assuming there is no redundancy as every locus can increase fitness and no locus can affect the potential of another locus to increase fitness. On the other hand, quantitative genetic models implicitly assume complete redundancy as the loci driving trait variation have interchangeable effects. Given that differences in redundancy result in quite different evolutionary dynamics (*Hollinger et al., 2019*; *Yeaman, 2022*), it is important to re-evaluate the behaviour of theoretical models in light of what we can learn about redundancy from empirical data. Our results suggest that adaptation is a complex process that does not map cleanly onto the assumptions of either approach. At least some component of trait variation experiences sufficiently strong selection and has low enough redundancy to drive the significant signatures of repeatability above the random null that we observe here. On the other hand, most of the strongest signatures of association are specific to a single species, implying high redundancy for most of the genome, even if there are still some detectable signatures of repeatability. Such observations may be consistent with classifying loci into 'core' and 'peripheral' sets similar to the omnigenic model, but would involve classification based on redundancy rather than gene expression, as advocated by Boyle et al. (2017). Theoretical models adopting a two-class representation of the genetic basis of trait variation may provide more realism than models assuming a simpler distribution, and perhaps yield new insights about qualitatively different dynamical behaviour. Given the difficulty of rigorously quantifying the contribution of alleles of small effect and loci contributing to local adaptation in only one species, it remains an open question what proportion of adaptive response is governed by loci with dynamical behaviour that follow the infinitesimal assumption. This could perhaps be quantified explicitly by decomposing heritabilities for traits driving local adaptation, as has been done for standing variation in humans (*Visscher et al., 2017*), but this would require a considerable increase in sample size as methods such as GCTA (*Yang et al., 2011*) do not yield accurate estimates at the sample sizes used here. Future work could perhaps resolve this question using multi-generational crossing designs to break up the LD that tends to accompany local adaptation, allowing a more accurate parsing of the effect sizes of genomic regions and the contribution by repeated vs. redundant components of adaptive trait variation.

## Materials and methods

### Data collection, common garden, and phenotyping

Seed samples were collected from 151 wild sunflower populations covering most of their native distributions during the summer of 2015 (*H. annuus*: 61 populations for GWAS and 71 populations for GEA; *H. petiolaris fallax*: 23 populations; *H. petiolaris petiolaris:* 18 populations; and *H. argophyllus*: 30 populations, *Figure 1* and *Supplementary file 1*). Seeds from 10 additional populations of *H. annuus* for the GEA analysis had been previously collected in the summer of 2011. Sample seeds were obtained from randomly chosen mothers and were first germinated in a greenhouse for 2 wk, later moved to an open-sided greenhouse for acclimation. Phenotypic data were collected throughout the growing season, as detailed in *Supplementary file 1*. Extensive records of developmental and morphological traits throughout the growth of the plants including leaves, stem, and seeds were collected and digitally imaged to extract relevant phenotypic data.

### Similarity in phenotype–environment correlation (SIPEC)

Locally adapted traits tend to exhibit strong correlations between environment and common-garden phenotype. To estimate which environmental variables are driving local adaptation for the same phenotype in pairs of species, we calculate an index of the similarity in phenotype–environment correlation, SIPEC = $(|r_1| + |r_2|) |r_1| |r_2|$ where $r_1$ and $r_2$ are the Pearson correlations between the environment and phenotype in the first and second species, respectively. This SIPEC index is maximized when the correlation between a phenotype and an environmental variable is large in both species regardless of the direction, so it does not differentiate phenotypically convergent vs. divergent patterns of local

adaptation (e.g. increasing temperature causing an increase in flowering time in one species and a decrease in the other), and provides a means of estimating the relative importance of an environment driving local adaptation in both species in a similar way across all measured phenotypes. To account for non-independence among traits, for each pair of species we fit a PCA to all measured phenotypes and use the principal components that collectively explain 95% of the variance, and calculate SIPEC on the correlations of each of these axes with environment. For comparisons including *H. argophyllus*, 95% of the variance was typically explained by 8–10 PC axes (out of 28 or 29 phenotypes), whereas for comparisons among other taxa this included 21 or 22 PC axes (out of 65 or 66 phenotypes). We then report the maximum and mean value of SIPEC for each environmental variable across these phenotypic PCA axes.

## Tests of SNP association with environment (GEA) and phenotype (GWAS)

A total of 39 environmental variables (21 climatic variables, 3 geographic variables, 15 soil variables; *Supplementary file 1*) were used for the genotype–environment association analysis (GEA). We refer to the climatic, soil, and geographic variables collectively as the 'environmental variables' for simplicity. Soil data were collected by taking 3–5 soil samples collected at each population from across the area in which seeds were collected and submitted to Midwest Laboratories Inc (Omaha, NE) for analysis. Climate data for each population were collected over a 30-year period (1961–1990) from geographic coordinates of the locations where the samples were collected using the software package Climate NA (*Wang et al., 2012*). We used the package BayPass version 2.1 (*Gautier, 2015*), which provides a re-implementation of the Bayesian hierarchical model, and explicitly accounts for the covariance structure among the population allele frequencies that originate from the shared history of the populations through the estimation of the population covariance matrix. This renders the identification of SNPs subjected to selection less sensitive to the confounding effect of population demography (*Gunther and Coop, 2013*). Population structure was estimated by choosing a random and unlinked set of 10,000 SNPs and running BayPass under the core model (i.e. no covariates). Then the Bayes factors (BF) were calculated running BayPass under the STD covariate model to evaluate association of SNPs with environmental variables (i.e. adjusting for population structure). For each SNP, the Bayes factor (denoted $BF_{is}$ as in *Gautier, 2015*) was presented in deciban units (db) via the transformation $10 \log_{10} (BF)$. $BF_{is}$ relies on the importance sampling algorithm proposed by *Gunther and Coop, 2013* and uses MCMC samples obtained under the core model. To produce a narrower set of outlier loci, we followed the popular Jeffreys' rule (*Jeffreys, 1961*) that identified outlier loci with $BF \geq 10$. As genome scan methods that correct for population structure can remove some potential signals of local adaption when there is covariation between the demographic history of the species and the environmental variables or phenotypic traits of interest, we also calculated Spearman's rank correlation ($\rho$, uncorrected GEA) between population allele frequencies for each SNP and each environment variable.

GWAS analysis was performed on 86, 30, and 69 developmental and morphological traits in *H. annuus, H. argophyllus,* and *H. petiolaris*, respectively (*Supplementary file 1*). In total, 29 variables were measured in all three focal species and 39 were measured only in *H. annuus* and in *H. petiolaris* subspecies (*Supplementary file 1*). Seed and flower traits could not be collected for *H. argophyllus* since most plants of this species flowered very late in our common garden and failed to form fully developed inflorescences and set seeds before temperatures became too low for their survival. Population structure was controlled for in GWAS by including the first three principal components as covariates, as well as an identity-by-state (IBS) kinship matrix calculated by EMMAX (*Kang et al., 2010*). We ran each trait GWAS using EMMAX (v07Mar2010), as well as the EMMAX module in EasyGWAS (*Grimm et al., 2017*). For every SNP/peak above the Bonferroni significance threshold, candidate genes were selected within a 100 kb interval centred in the SNP with the lowest p-value, or within the boundaries of the GWAS peak, whichever was larger. All variants used for association were initially filtered for VQSR 90% tranche, and then further filtered to only include bi-allelic SNPs genotyped in ≥90% of samples and with a minor allele frequency ≥3%.

## Identification of top candidate windows

We calculated the bottom 0.05 quantiles for the p-values from association tests, Spearman's rank correlation (uncorrected GEA), and GWAS (corrected and uncorrected), yielding two 5% cutoffs. For

each environmental and phenotypic variable, we identified all outlier SNPs as those that fell below the respective 5% cutoff. For BayPass, we considered SNPs with BF ≥ 10 as outlier SNPs. For each 5 kb window that we defined across the whole genome, we counted the number of outlier SNPs ($a$) and the total number of SNPs ($n$). To identify top candidate windows for each variable, we compared the number of outlier SNPs per each 5 kb window to the 0.9999 quantile of the binomial expectation where the expected frequency of outlier SNPs per window is calculated as $\rho = \sum a_i/n_i$ (summation over all 5 kb windows), calculating $\rho$ independently for each environmental and phenotype variable and excluding windows with 0 outliers from the calculation of $\rho$ (as per *Yeaman et al., 2016*). We also calculated a top candidate index using the same approach to categorize outliers, obtaining a p-value for a binomial test for the number of SNPs per window given an expected proportion of outliers ($\rho$; this p-value is not exact due to non-independence of SNPs, so we refer to this as an index).

Identifying outlier SNPs detected by genome scans from genome-wide distribution without accounting for local recombination rate variation can promote false-positive signals in recombination cold spots (i.e. low recombination regions) and be overly conservative in recombination hot spots (i.e. high recombination regions) (*Booker et al., 2020*). Therefore, to account for local recombination rate variation, all genomic windows were binned by their estimated recombination rates into five equally sized bins (bin1: 0–20% quantile; bin2: 20–40% quantile; bin3: 40–60% quantile; bin4: 60–80% quantile; bin5: 80–100% quantile). For each recombination bin, we estimated expected frequency of SNPs per window ($\rho$) and calculated cutoff separately. Windows falling above the threshold were identified as top candidate windows.

## Genome-wide survey of repeatability (null-W test)

To explore repeatable genomic signatures of adaptation for each of the six pairwise contrasts among the four taxa (*Figure 1*), we used the method developed by *Yeaman et al., 2016* with some modifications. A common approach is to identify candidates for adaptation independently in each species and then examine the overlap between these lists; however, this approach is quite stringent and may miss many interesting signals. The null-W test is more sensitive as it takes the list of top candidates from one species and tests whether they tend to show more extreme signatures of association than expected by chance. The null-W test is especially favourable when LD increases divergence of SNPs in tight linkage with causal SNPs but does not raise the test values enough for a window to be classified as an outlier according to the binomial test. For each top candidate window that we identified for each focal species in a pair, we refer to the same window in the other species as 'top candidate ortholog'. The null distribution for each focal species and variable was constructed by randomly sampling 10,000 background SNPs from non-top candidate ortholog windows. For each non-top candidate window, we then estimated the test statistic ($W$) for the Wilcoxon signed-rank test vs. the 10,000 background SNPs. This resulted in a null distribution representing the differences between the 10,000 background SNPs and the non-top candidate ortholog windows. These were then standardized to $Z$-scores using the method in *Whitlock and Schluter, 2020*:

$$Z = \frac{2W - n_1 n_2}{\sqrt{n_1 n_2 \left(n_1 + n_2 + 1\right)/3}} \tag{1}$$

where $n_1$ and $n_2$ are the sample sizes being compared. In order to control for heterogeneity in recombination rate and its possible effects on the null distribution, we estimated null distribution for each recombination bin separately (five in total, see above). We then compared the p-values and BFs for each focal top candidate window to the 10,000 background SNPs, calculating the W statistics and converting into a Z-score. Empirical p-values were then calculated by comparing the Z-score for each top candidate window to the null distribution. When individual windows had values of W that exceeded the bounds of the null distribution, their empirical p-value was set to the reciprocal of the number of genes in the null distribution. For each species pair, we refer to the windows identified as significant by this test as 'windows of repeated association' or WRAs.

## Linkage disequilibrium and detection of clusters of repeated association

LD among adjacent genomic windows can result in statistical non-independence and similar GEA/GWAS signatures across many windows. To identify the most significant WRAs and group neighbouring windows with similar signatures of repeatability into a single CRA, we used the following approach in each pairwise contrast, and for each environmental variable and phenotype: for each variable, empirical p-values for all WRAs were converted to $q$-values to adjust for false discovery, then beginning with the first significant WRA along the chromosome (i.e., with $q < 0.05$), we compared it to the next closest significant WRA by calculating the squared Pearson correlation coefficient ($r^2$) on the allele frequencies across all pairs of SNPs and compared this estimate to a null distribution. To construct the null distribution, we generated a distribution of LD measurements between 10,000 randomly chosen windows with the same physical distance as between two significant WRAs (excluding all WRAs from the null distribution). If the $r^2$ between the two neighbouring significant WRAs was greater than the 95th percentile of the null distribution, we clustered these two windows together. This process was repeated successively with the next closest neighbouring significant WRA, walking out along the chromosome until one of two stopping criteria was reached: (1) the LD between the last two windows did not exceed the 95% of the tail distribution or (2) the distance between the initial window and the current window next to it was larger than 1 cM, based on the linkage map from *Todesco et al., 2020*. When the first round of clustering stopped due to either of these two criteria, all the clustered windows were removed from the dataset and the process started with the second smallest empirical p-value. By doing this way, each significant WRA will only appear in a single CRA.

## Estimating an index of shared standing variation

As genotype calling was conducted separately for each species (due to computational concerns), we estimated the amount of shared standing variation based on counts of shared vs. non-shared SNPs. If two species are evolving independently, the number of shared SNPs should follow a hypergeometric distribution, so we used an approach similar to the C-scores (*Yeaman et al., 2018*) to calculate the difference between the observed number of shared SNPs and the expectation, scaled by the standard deviation of the hypergeometric. Because of noise and a relatively small number of SNPs per window, we applied this approach on a sliding-window basis, including the five flanking windows upstream and downstream of each focal window the calculation of its index of shared standing variation.

## Correspondence between regions of repeated association and chromosomal rearrangements

To assess the extent of overlap between regions of the genome with repeated signatures of association and previously identified low-recombination haploblocks, we used two approaches. Firstly, for each pairwise species contrast and variable, we constructed a contingency table for the number of top candidates that were significant WRAs vs. non-significant WRAs (by the null-W test), and that did vs. did not fall within a haploblock. We calculated the Pearson's $\chi^2$ statistic on this table and then permuted the location of haploblocks throughout the genome to construct a null distribution of $\chi^2$ statistics, and calculated the p-value as the proportion of the null that exceeded the observed $\chi^2$ statistic (to account for non-independence of nearby WRAs), which is presented in *Figure 5*. Secondly, we compared observations of the length and number of overlapping regions to expectations based on a randomization approach. For each pair of species, we kept the position of each CRA constant and randomized the position of haploblocks 10,000 times to build a null distribution. By keeping the position of the CRAs constant, we maintained the architecture of adaptation independent from chromosomal rearrangements. We assessed significance by testing whether the observed overlaps between CRAs and haploblocks were more extreme than the 95th percentile of the tail of null distribution, which is presented in *Appendix 1—figure 7*. Details about identifying chromosomal rearrangements can be found in *Todesco et al., 2020*.

## Identifying repeated signatures of association across all taxa

As a complement to the pairwise analysis, we used PicMin (Booker et al. 2022a) to identify windows of the genome with strong signatures of association in multiple (sub-) species. For each environmental variable, association signatures for each window are ranked genome-wide and PicMin identifies

windows with extreme ranks in multiple species. We ran the analysis once with each of the two 3-way comparisons involving one *petiolaris* subspecies (i.e. *H. annuus, H. argophyllus,* and either *H. pet. petiolaris* or *H. pet. fallax*) to control for repeatability arising from similar patterns in the two *petiolaris* subspecies (due to high introgression/shared standing variation).

## Evaluating genotypic redundancy using C-scores

Genome-wide quantification of the repeatability of association statistics provides insights into the amount of genotypic redundancy underlying a trait or environmental adaptation, and can be assessed using the C-score approach (*Yeaman et al., 2018*). Briefly, for a given trait or environment, the set of association test scores within each species is classified into 'associated' or 'non-associated' using the binomial top candidate approach. For a given pair of species ($i$ = 1, 2), the observed number of associated windows in each species ($a_1$, $a_2$) can be compared to the number of windows that are associated in both species ($a_b$) and the total number of windows being analysed ($a_x$). Under a null hypothesis where all windows in the genome have equal potential to be involved in adaptation (i.e. associated with the trait or environment), the expected number of windows associated in both species will be described by a hypergeometric distribution, with the expectation $\bar{a_b} = a_1 a_2 / a_x$. The difference between the observed and expected amount of overlap in association scores can be quantified as a C-score by scaling the difference by the standard deviation of the hypergeometric (i.e. a C-score of 2 means that the observed amount of overlap is 2 standard deviations above the expectation under randomness).

We assess the C-scores for each phenotype and environment trait by classifying the top 0.5% of all 5 kb windows within each species as 'associated' ($a_i$; based on the binomial top candidate index), and begin by calculating the C-score obtained when $a_x$ is set to the union of $a_i$ across all four focal species/sub-species (i.e. only those windows associated in at least one species are included in $a_x$, a number much lower than the total number of windows in the genome). This yields a negative C-score as a random draw from such a small number of windows tends to yield many more overlapping associations by chance than the observation. When the C-score = 0, it means that the observed overlap between the pair of taxa being considered is consistent with a random draw of their respective 'associated' windows from an overall pool of $a_x$ windows. Thus, by adding rows to the matrix with 'non-associated' scores for all four species/sub-species until finding the matrix that yields C-score = 0, we can estimate the effective number of loci that contributed to variation for the trait or environment being considered ($L_{eff}$), under the assumption that all such $a_x = L_{eff}$ windows had an equal chance of contributing to the associations.

To run the C-score analysis on LD-clustered windows, we ran the following algorithm for each trait/environmental variable: for each species, we identified all CRAs that also had a top candidate index in top 0.5%. In most cases, a large cluster with associated windows in one species corresponds to either no clusters or a small cluster in another species. To harmonize cluster boundaries across species, we bin any overlapping clusters together to their maximum extent and take the average top candidate index for each cluster in each species as either (a) the mean across all windows that were 'associated' in that species or (b) the average across all windows associated in any species, if no windows were associated in that species. This yields a matrix that can be submitted to the hypergeometric C-score analysis.

## Acknowledgements

The authors thank the Yeaman and Peichel labs for comments. Computational support was provided by the Digital Resources Alliance of Canada, and funding was provided by Genome Canada and Genome BC (LSARP2014-223SUN), the International Consortium for Sunflower Genomic Resources, Alberta Innovates, and NSERC Discovery. GLO was funded by a Banting Postdoctoral Fellowship.

## Additional information

### Funding

| Funder | Grant reference number | Author |
|---|---|---|
| Genome Canada | LSARP2014-223SUN | Loren H Rieseberg<br>Sam Yeaman |
| Genome British Columbia | LSARP2014-223SUN | Loren H Rieseberg<br>Sam Yeaman |
| International Consortium for Sunflower Genomic Resources | | Loren H Rieseberg |
| Alberta Innovates | 212201729 | Sam Yeaman |
| Natural Sciences and Engineering Research Council of Canada | RGPIN/03950-2017 | Sam Yeaman |
| Canadian Institutes of Health Research | Banting Postdoctoral Fellowship | Gregory L Owens |
| Natural Sciences and Engineering Research Council of Canada | Banting Postdoctoral Fellowship | Gregory L Owens |
| Social Sciences and Humanities Research Council of Canada | Banting Postdoctoral Fellowship | Gregory L Owens |

The funders had no role in study design, data collection and interpretation, or the decision to submit the work for publication.

### Author contributions

Shaghayegh Soudi, Mojtaba Jahani, Resources, Formal analysis, Investigation, Visualization, Writing – original draft, Writing – review and editing; Marco Todesco, Gregory L Owens, Visualization, Formal analysis, Investigation, Methodology, Writing – review and editing; Natalia Bercovich, Resources, Visualization, Methodology; Loren H Rieseberg, Conceptualization, Supervision, Investigation, Project administration, Writing – review and editing; Sam Yeaman, Conceptualization, Resources, Formal analysis, Supervision, Investigation, Visualization, Methodology, Writing – original draft, Project administration, Writing – review and editing

### Author ORCIDs

Sam Yeaman (ID) https://orcid.org/0000-0002-1706-8699

Reviewer #1 (Public Review): https://doi.org/10.7554/eLife.88604.3.sa1
Reviewer #2 (Public Review): https://doi.org/10.7554/eLife.88604.3.sa2
Author Response https://doi.org/10.7554/eLife.88604.3.sa3

---

## Additional files

### Supplementary files

• Supplementary file 1. Tables showing phenotypes and environmental variables for all sampled individuals and populations, as well as legends for details associated with each variable (reproduced from *Todesco et al., 2020*).

• MDAR checklist

### Data availability

All scripts used for analysis and figures are deposited in a Dryad Digital Repository. Raw sequence data are deposited under NCBI BioProject accessions PRJNA532579, PRJNA398560, and PRJNA564337, as described in *Todesco et al., 2020*.

The following dataset was generated:

| Author(s) | Year | Dataset title | Dataset URL | Database and Identifier |
|---|---|---|---|---|
| Sam Y, Shaghayegh S, Mojtaba J | 2023 | Data from: Repeatability of adaptation in sunflowers reveals that genomic regions harbouring inversions also drive adaptation in species lacking an inversion | https://doi.org/10.5061/dryad.wpzgmsbtd | Dryad Digital Repository, 10.5061/dryad.wpzgmsbtd |

The following previously published datasets were used:

| Author(s) | Year | Dataset title | Dataset URL | Database and Identifier |
|---|---|---|---|---|
| Todesco M | 2020 | Wild Helianthus GWAS and GEA | https://www.ncbi.nlm.nih.gov/bioproject/?term=PRJNA532579 | NCBI BioProject, PRJNA532579 |
| Todesco M | 2020 | Wild and Weedy Helianthus annuus whole genome resequencing | https://www.ncbi.nlm.nih.gov/bioproject/?term=PRJNA398560 | NCBI BioProject, PRJNA398560 |
| Todesco M | 2020 | Hi-C sequencing of wild sunflowers | https://www.ncbi.nlm.nih.gov/bioproject/?term=PRJNA564337 | NCBI BioProject, PRJNA564337 |

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

# Appendix 1

## Additional methods

### Whole-genome re-sequence data alignment and variant calling

DNA was extracted from leaf tissues (these are the same individuals examined in the previous study, see *Todesco et al., 2020* for details). Briefly, all libraries were sequenced at the Genome Quebéc Innovation Center on HiSeq2500, HiSeq4000, and HiSeqX instruments to produce paired end, 150 bp reads (Illumina, San Diego, CA). Libraries with a low number of reads were re-sequenced to increase genome coverage. After quality filtering (see below), a total of 60.7 billion read pairs were obtained. Illumina adapters and poor quality reads were hard-clipped using Trimmomatic (v0.36) (*Bolger et al., 2014*). Reads were then aligned to the *H. annuus* XRQv1 genome (HanXRQr1.0-20151230; *Badouin et al., 2017*) using NextGenMap (v0.5.3; *Sedlazeck et al., 2013*). PCR duplicates were marked and removed using picard MarkDuplicates 2.9.3. Genomic regions containing transposable elements (~3/4 of the sunflower genome) were excluded to reduce computational time and improve variant quality. Genotyping for each species was performed independently as joint genotyping on the whole ensemble of samples was computational impractical. GATK's VariantRecalibrator (v4.0.1.2; *Van der Auwera and O'Connor, 2000*), which filters variants in the call set according to a machine learning model inferred from a small set of 'true' variants, was used to remove low-quality calls and produce a dataset of a more manageable size. In the absence of an externally validated set of known sunflower variants to use as calibration, we computed a stringently filtered set from top-N samples with highest sequencing coverage for each species (N = 67 for cultivated sunflower and N = 20 for wild sunflower species). The stringency of the algorithm in classifying true/false variants was adjusted by comparing variant sets produced for different parameter values (tranche 100.0, 99.0, 90.0, 70.0, and 50.0). For each cohort, results for tranche = 90.0 were chosen for downstream analysis, based on heuristics: the number of novel SNPs identified, and improvements to the transition/transversion ratio (towards GATK's default target of 2.15).

### Remapping sites to the HA412-HO reference genome

As described with details by *Todesco et al., 2020*, haploblock analysis highlighted contig ordering issues with the XRQv1 reference assembly (see below). To overcome this, all sites were transferred to a new reference, HA412-HOv2, which used Hi-C for contig and scaffold ordering (*Belton et al., 2012*; *Marie-Nelly et al., 2014*). To do this, the 200 bp of reference sequence flanking each site in XRQv1 were extracted and aligned to HA412-HOv2 using BWA (*Li, 2013*). These alignments were filtered for mapping quality >40 and the HA412-HOv2 position for the variant site was extracted. Since all remapped sites were not in repetitive regions and had passed VQSR filtering, remapping success rate was high (96–98%). Whenever mapping suggested two different variants on the XRQv1 genome were in the same position on the HA412-HOv2 genome, likely due indels and imprecise alignment, one site was shifted by 1 bp so that they did not overlap. Remapping was preferred to de novo read alignment and variant calling against the HA412-HOv2 assembly because of the prohibitive amount of computational time that would have required.

### Gene Ontology enrichment analysis of regions with repeated association

Genes that overlapped with CRAs associated with environmental and phenotypic variables were screened for enrichment of Gene Ontology (GO) terms. GO annotations for *Arabidopsis thaliana* genes from the TAIR database were mapped onto their sunflower homologs and a custom database of sunflower GO annotations was constructed. The R package TOPGO (*Alexa and Rahnenfuhrer, 2023*) was used to analyse the set of candidate genes to determine which categories were most overrepresented. Significance for each individual GO identifier was computed with Fisher's exact test and significant GO terms were identified at an FDR of 1%. GO functional enrichment analysis was performed in the categories biological process (BP), cellular component (CC), and molecular function (MF).

## Additional results

### GO-enrichment analysis

To investigate the functional associations of genes overlapping with the WRA, we performed GO analysis using TopGO package in R. GO terms with corrected p-values<0.05 were considered significantly enriched. *Appendix 1—figures 9 and 10* provide a list of GO terms that are over-represented in our gene set. GO components and processes associated with membrane assembly,

transport through the endomembrane system, mRNA, and growth are significantly enriched in this analysis.

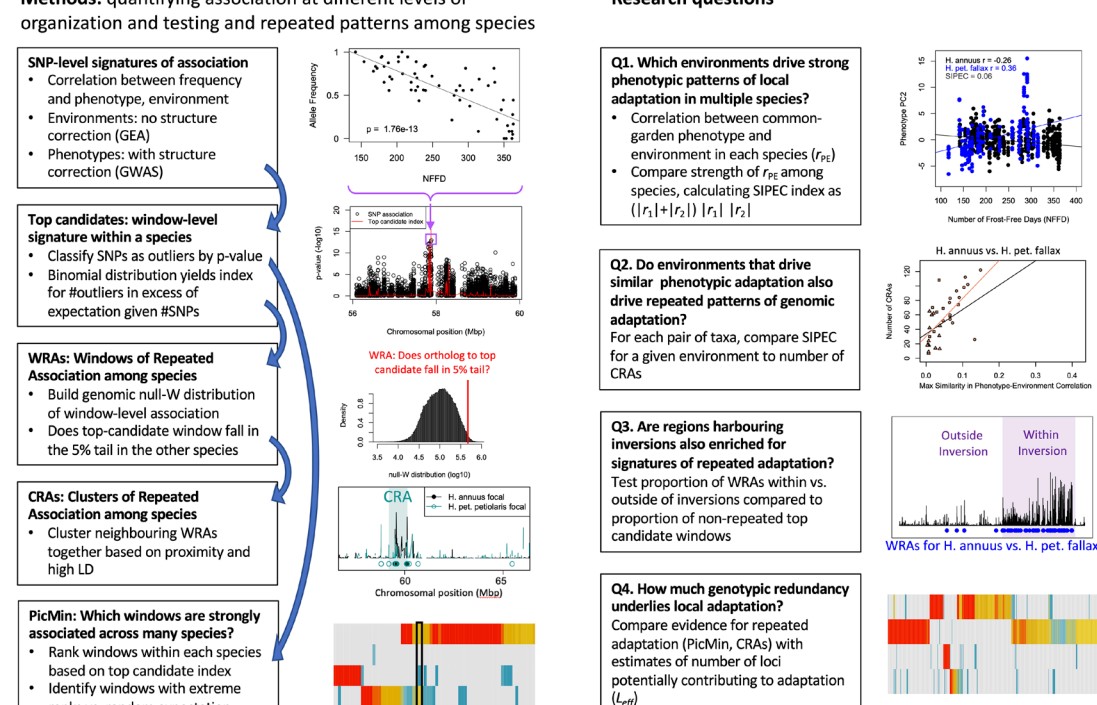

**Appendix 1—figure 1.** Schematic overview of methods and primary research questions.

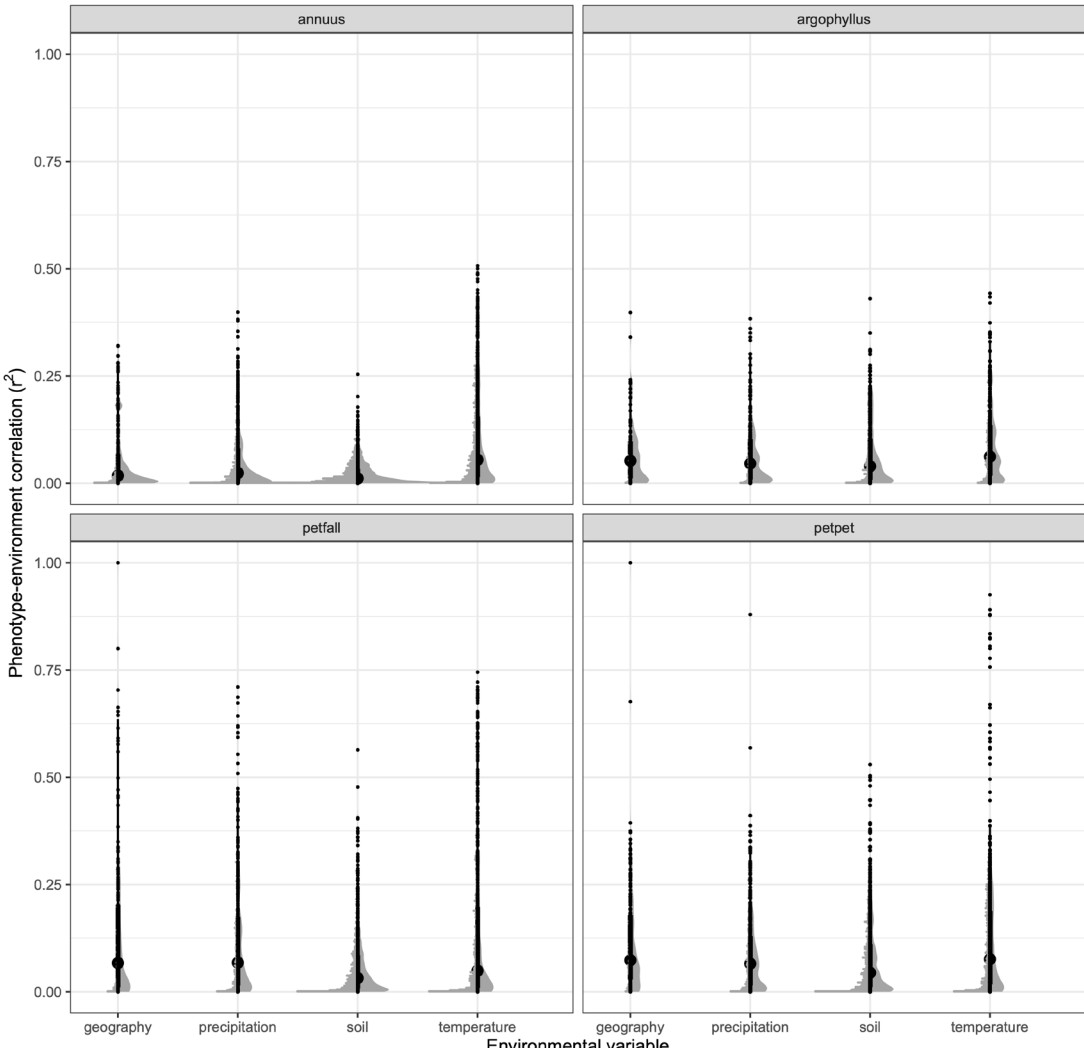

**Appendix 1—figure 2.** Strength of phenotype–environment correlations across all traits for four different types of environmental variables in each of the sunflower species and subspecies. Black points show individual values, and grey points show binned density.

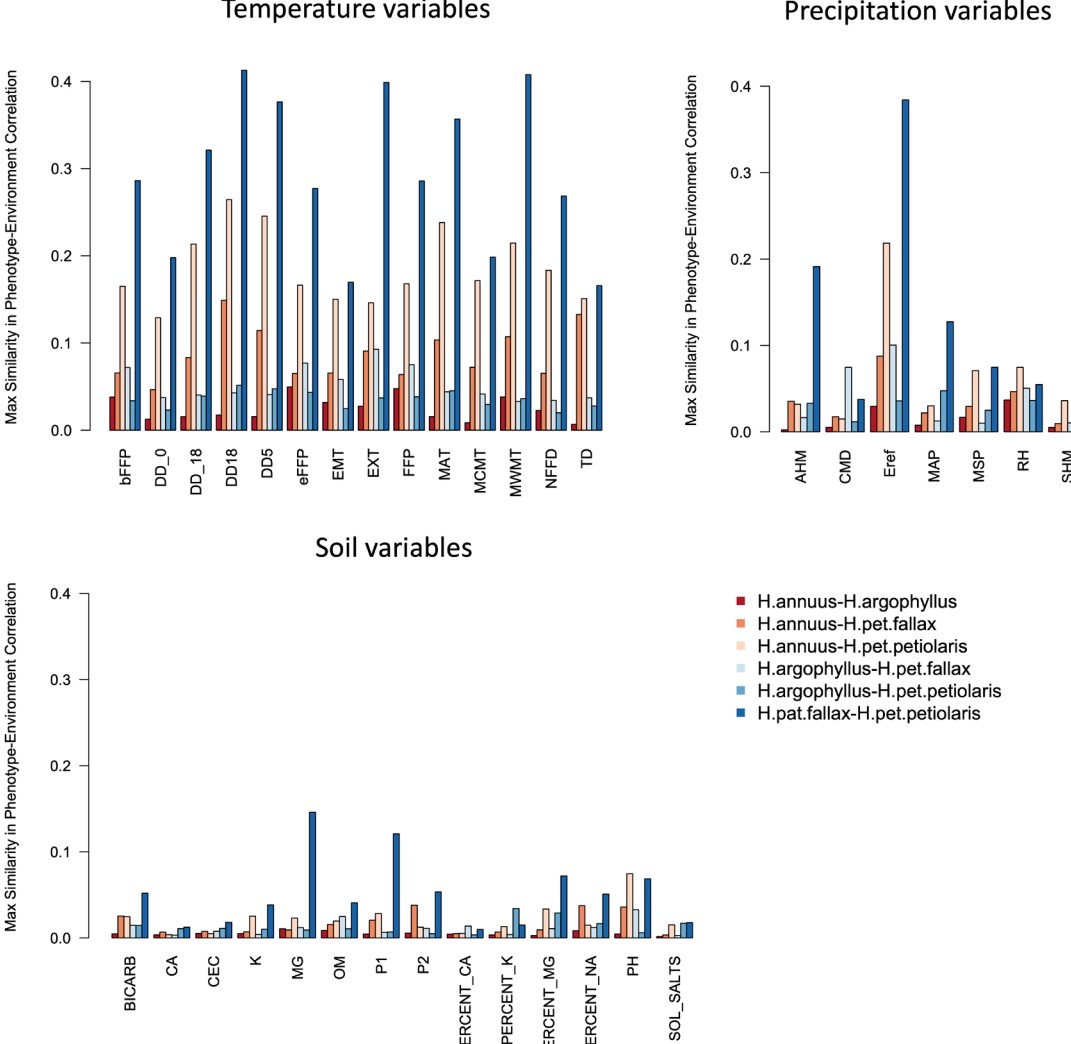

**Appendix 1—figure 3.** Maximum similarity in phenotype–environment correlation (SIPEC) for pairs of taxa, across soil-, temperature-, and precipitation-related environmental variables. The maximum value of SIPEC for each environment is calculated across all phenotypic principal component analysis (PCA) axes that cumulatively explain 95% of the variance.

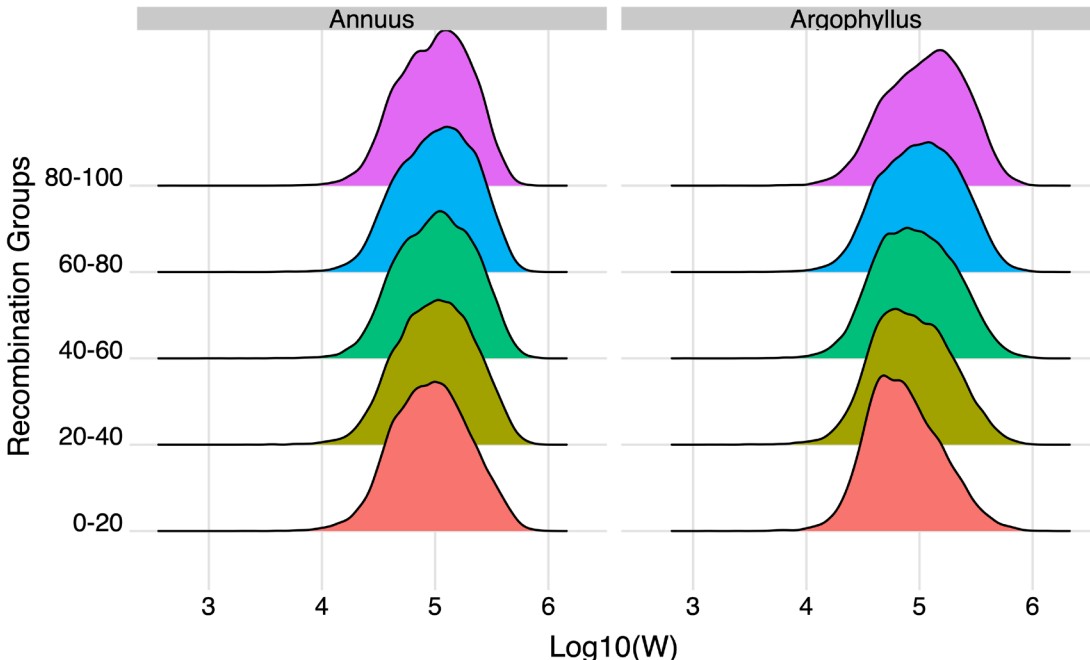

**Appendix 1—figure 4.** The effect of recombination rate on width of the null-W distribution for the number of frost-free days (NFFD) variable for *Helianthus annuus* and *H. argophyllus*. Recombination bins represent the 0th–20th percentile, 20th–40th percentile, etc.

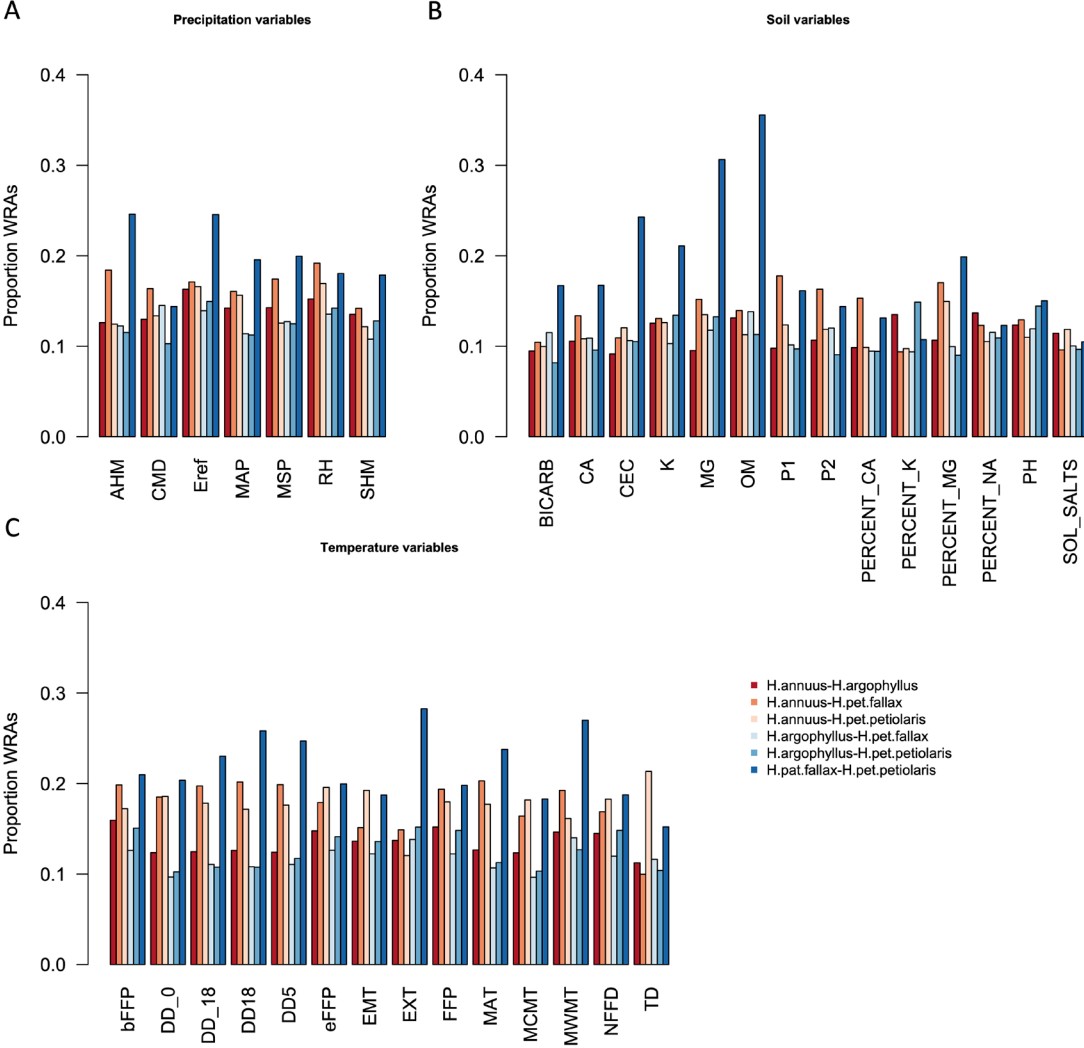

**Appendix 1—figure 5.** Proportion of top candidate windows that are significant hits under the null-W test (windows of repeated association), for pairs of taxa, across precipitation (**A**), soil (**B**), and temperature (**C**) environmental variables.

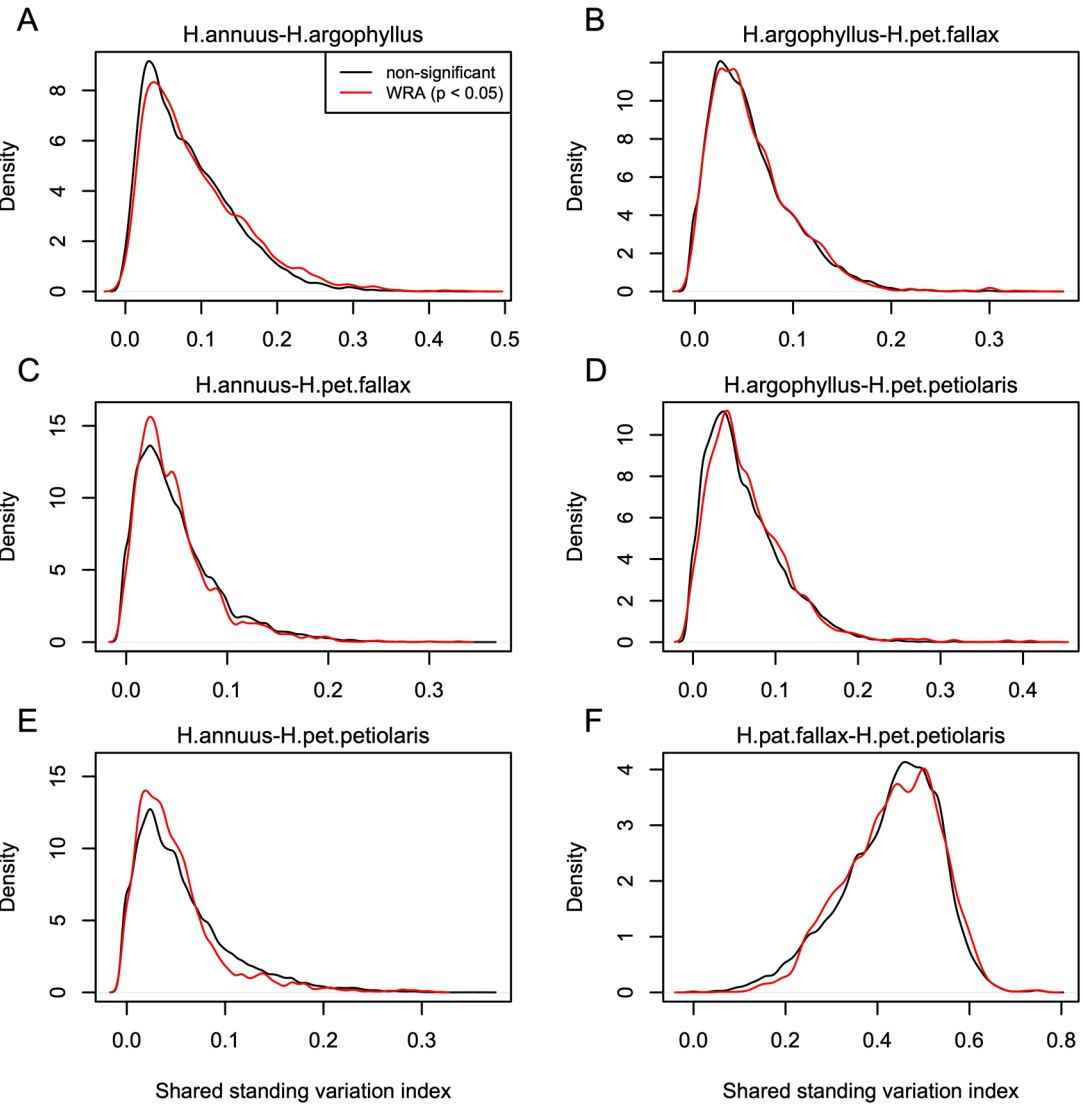

**Appendix 1—figure 6.** Index of shared standing variation for windows of repeated association (WRAs) vs. top candidates that were not significant under the null-W test. The index of shared standing variation reflects the proportion of SNPs that are shared vs. non-shared among species and provides an indicator of the likely extent of introgression, which does not appear to differ substantially among the windows of the genome with significant (p<0.05; red lines) vs. non-significant (p>0.05; black lines) null-W test results. Panels **A-F** show the values for each of the six pairwise comparisons among species, as indicated above each panel.

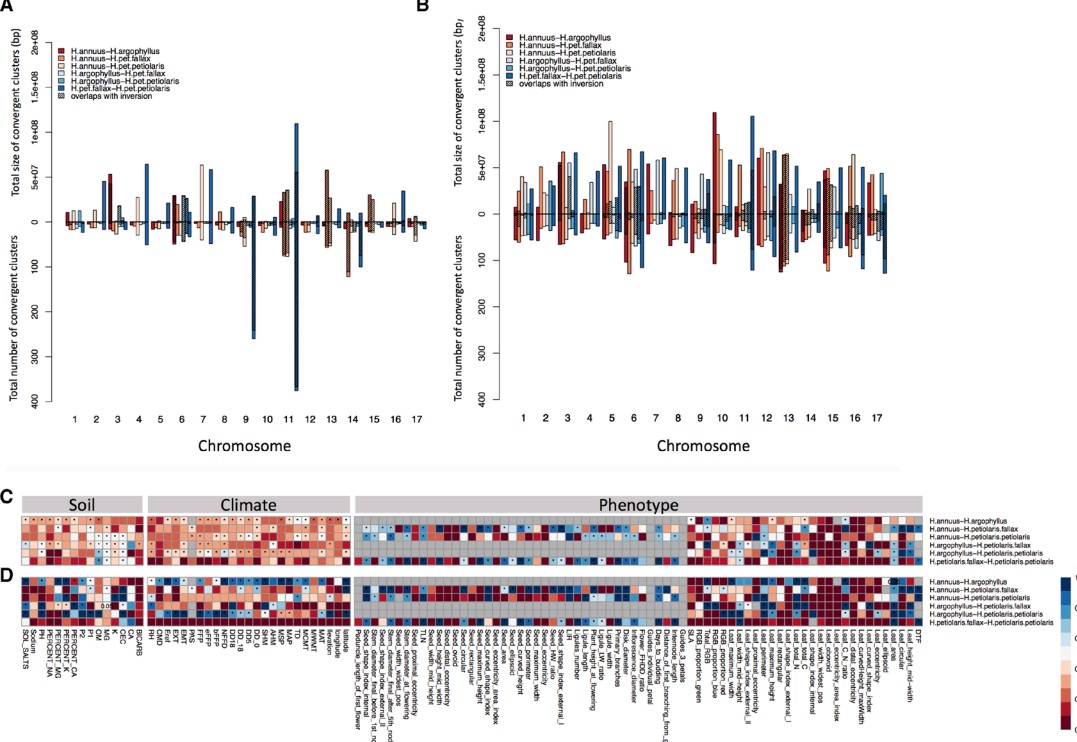

**Appendix 1—figure 7.** Number and size of clusters of repeated association (CRAs) and their overlap with haploblocks. Total size and total number of CRAs detected among six studied pairs on each linkage group across all phenotypes by Genome-Wide Association (GWAS) (**A**), and Genotype-Environment Association (GEA) (**B**). Hatching areas indicate the total size and number of clusters residing within chromosomal rearrangements. Heat maps present proportion of CRAs by number (**C**) and size (**D**) per each phenotype variable and environment variable overlapping with chromosomal rearrangements. Stars in indicate overlaps between CRAs and haploblocks happen significantly different from chance (p-value ≤ 0.05). Grey cells in the heat maps indicate no data is available for that comparison and variable.

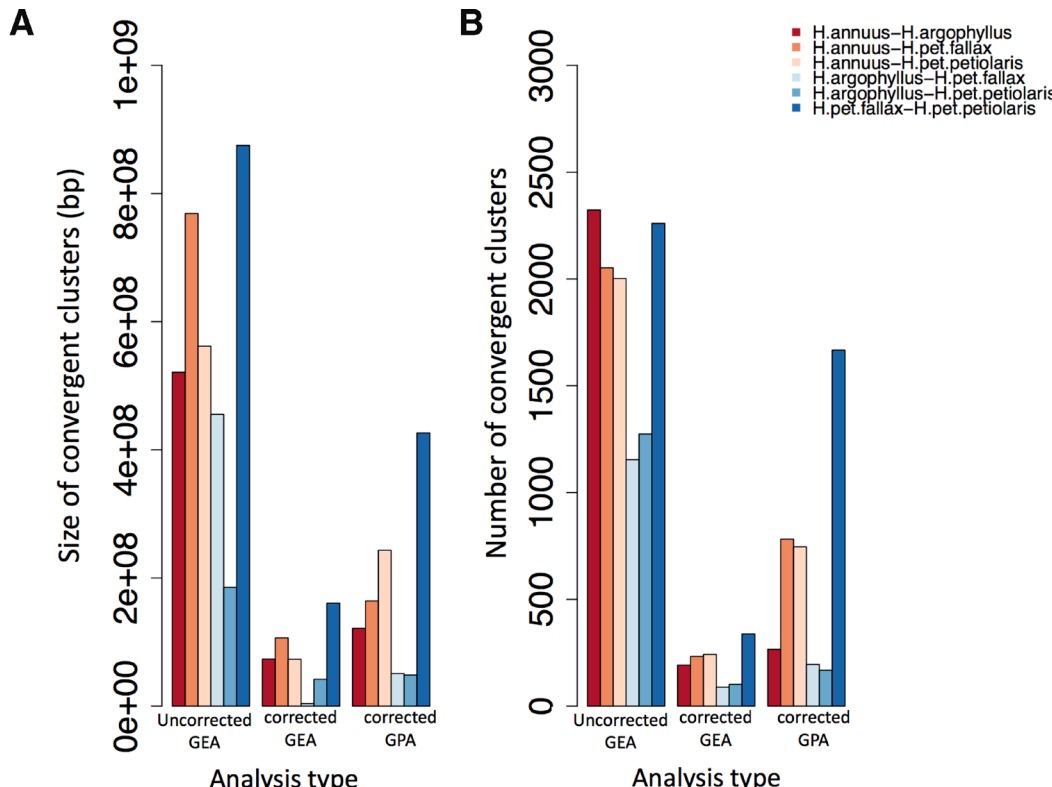

**Appendix 1—figure 8.** Total size and number of convergent clusters in different pairs for each analysis type. The total size of convergent clusters (**A**) and total number of convergent clusters (**B**) identified among different pairs surveyed in the present study using association genetic approaches that corrected population structure versus those that did not correct across all environmental variables (GEA) and corrected GWAS.

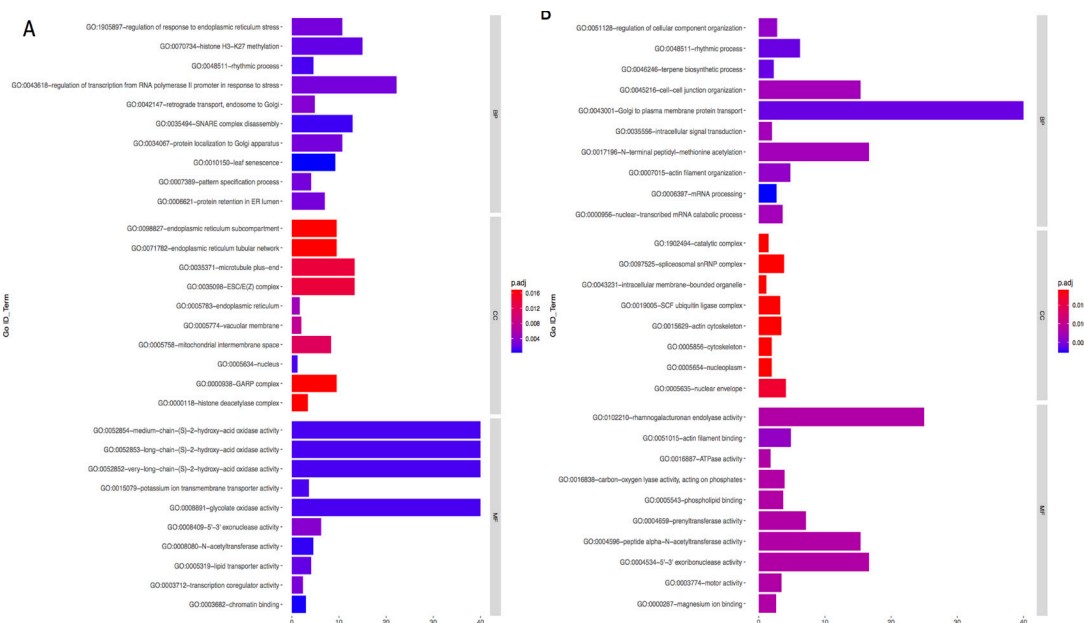

**Appendix 1—figure 9.** Bar graph of Gene Ontology (GO) enrichment analysis for phenotype (**A**) and precipitation-related variables (**B**). Bar plot depicts the significant enriched GO terms within categories: biological process, cellular component, and molecular function. Y-axis represents the GO term, and the X-axis represents the enrichment significance, respectively.

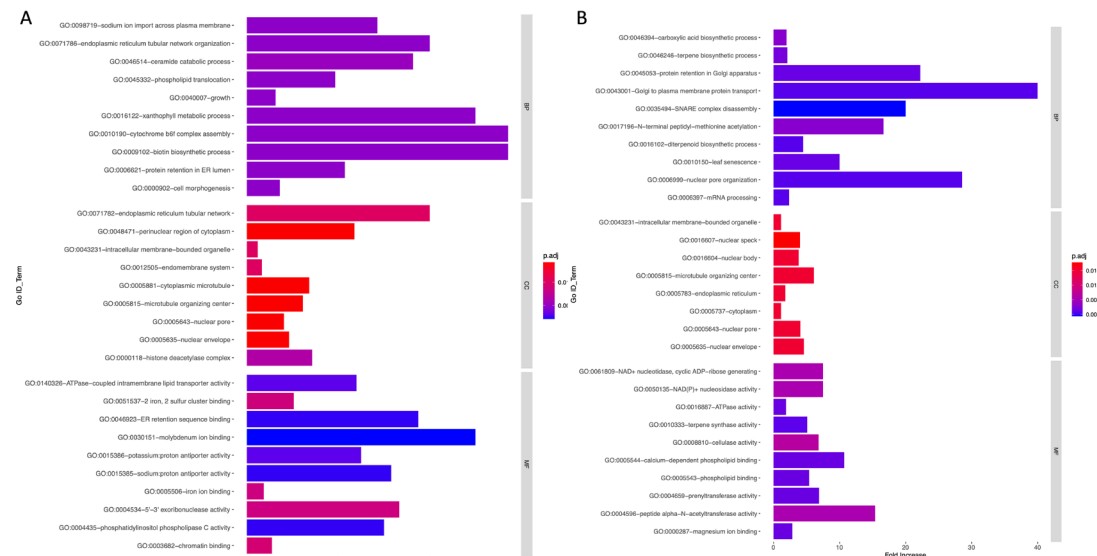

**Appendix 1—figure 10.** Bar graph of Gene Ontology (GO) enrichment analysis for soil (**A**) and temperature-related variables (**B**). Bar plot depicts the significant enriched GO terms within categories: biological process, cellular component, and molecular function. Y-axis represents the GO term, and the X-axis represents the enrichment significance.

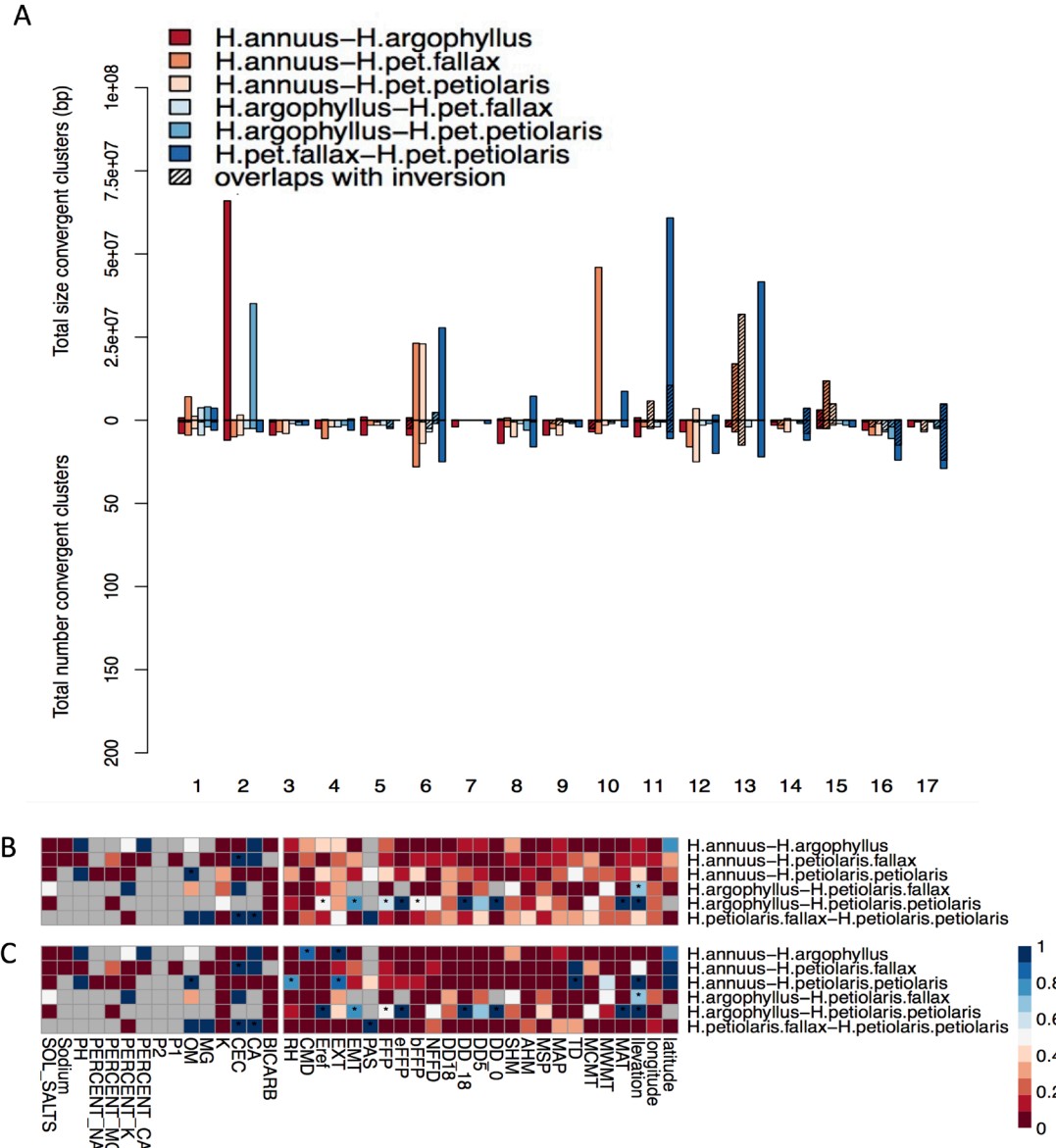

**Appendix 1—figure 11.** Effect of structure correction on number and size of clusters of repeated association (CRAs) and their overlaps with inversions. Total size and total number of CRAs detected among six studied pairs on each linkage group across all environmental variables by corrected GEA (**A**). Hatching areas indicate the total size and number of clusters residing within haploblocks. Heat maps present proportion of CRAs overlapping with haploblocks by number (**B**) and size (**C**) for each phenotype and environmental variable (climate and soil). Stars indicate cases where observed overlaps between CRAs and haploblocks happen significantly more than expected by chance (p-value ≤ 0.05). Grey cells in the heat maps indicate no data is available for that comparison and variable.

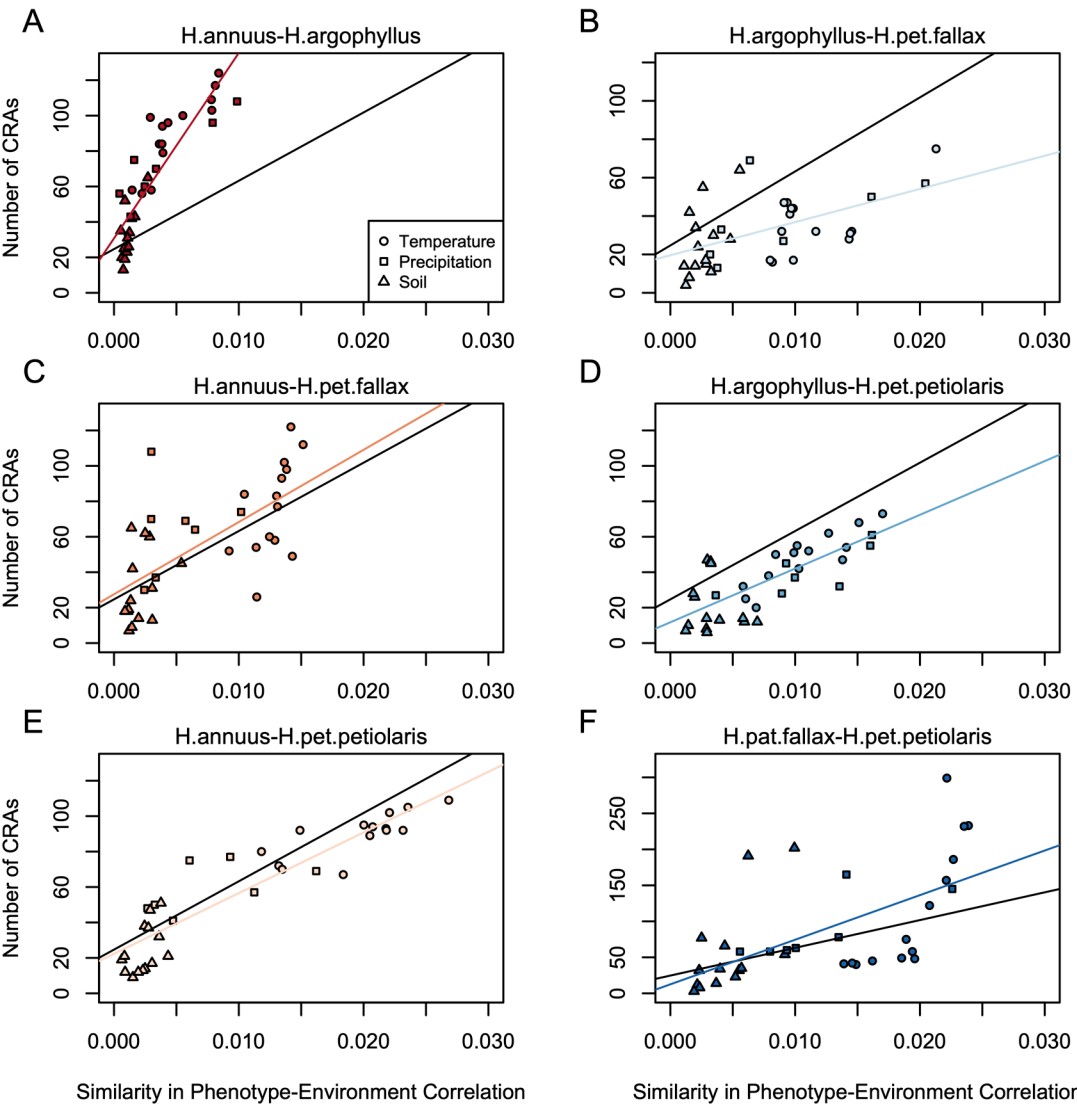

**Appendix 1—figure 12.** Relationship between mean similarity in phenotype-environment correlation (SIPEC) and number of clusters of repeated association (CRAs). SIPEC was calculated for each phenotypic principal component analysis (PCA) axis with the mean calculated across the axes that cumulatively explain 95% of the phenotypic variance for each environment. Each panel (**A-F**) shows a comparison between a pair of species indicated above, and includes both a linear model fit to the data within the panel (coloured lines), and a linear model fit to all data simultaneously (black lines) for comparison. Note that because environmental variables are correlated, these points are not independent and therefore represent a source of pseudoreplication, preventing formal statistical tests of this relationship.

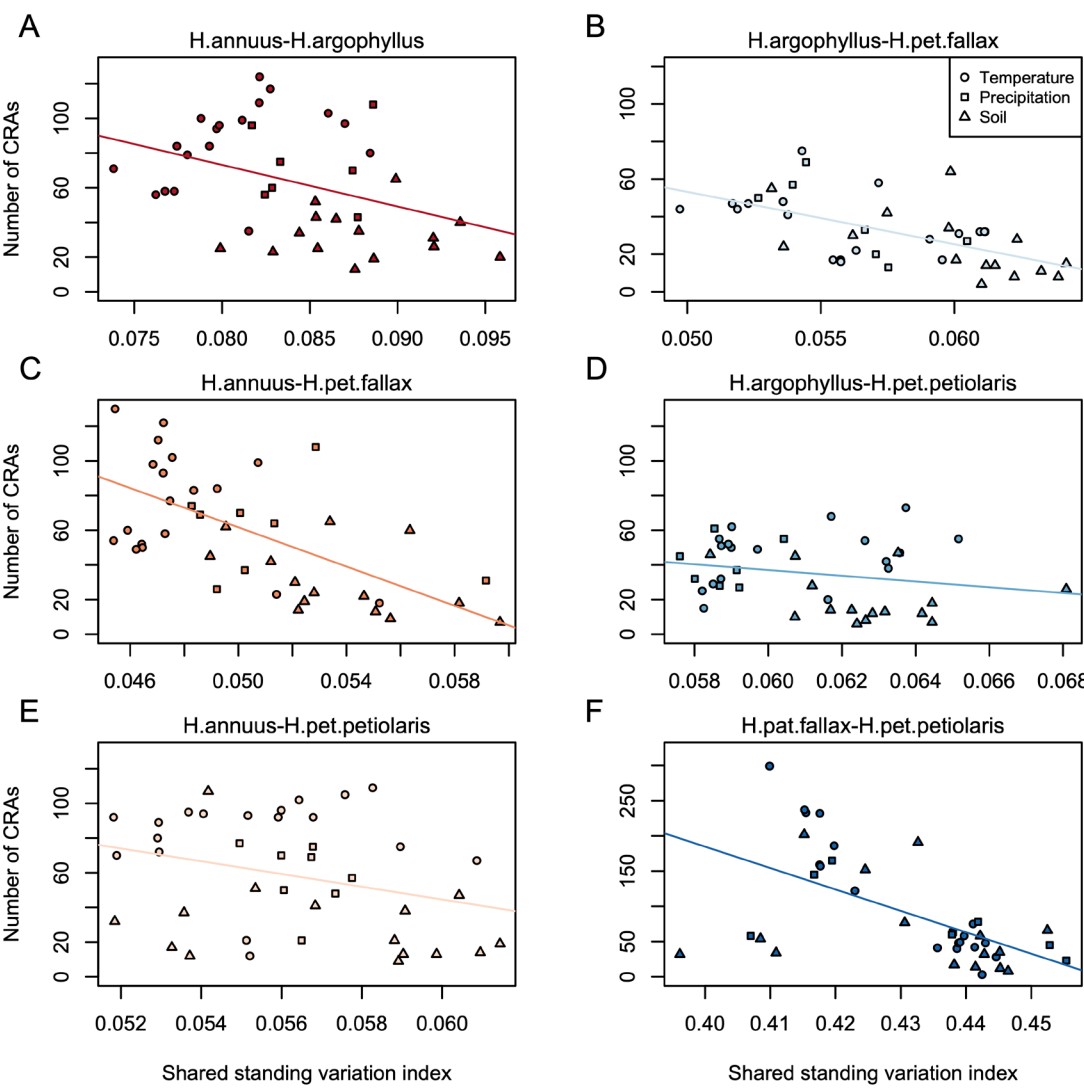

**Appendix 1—figure 13.** Relationship between index of shared standing variation and number of clusters of repeated association (CRAs). Panels **A-F** show the values for each of the six pairwise comparisons among species, as indicated above each panel, with lines showing linear model fits for data within each panel.

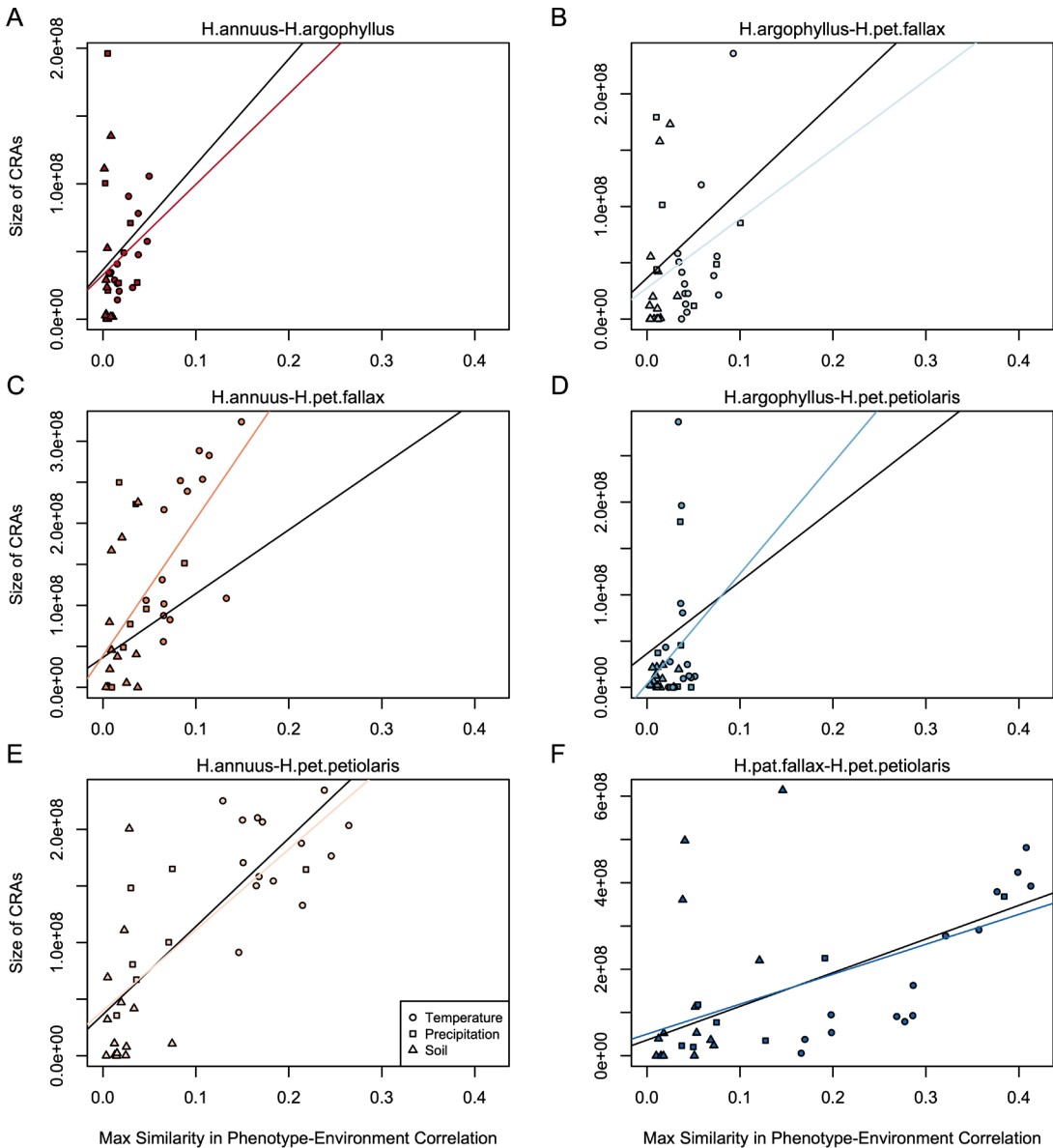

**Appendix 1—figure 14.** Relationship between mean similarity in phenotype–environment correlation (SIPEC) and size of clusters of repeated association (CRAs). SIPEC was calculated for each phenotypic principal component analysis (PCA) axis with the mean calculated across the axes that cumulatively explain 95% of the phenotypic variance for each environment. Each panel (**A-F**) shows a comparison between a pair of species indicated above, and includes both a linear model fit to the data within the panel (coloured lines), and a linear model fit to all data simultaneously (black lines) for comparison. Note that because environmental variables are correlated, these points are not independent and therefore represent a source of pseudoreplication, preventing formal statistical tests of this relationship.

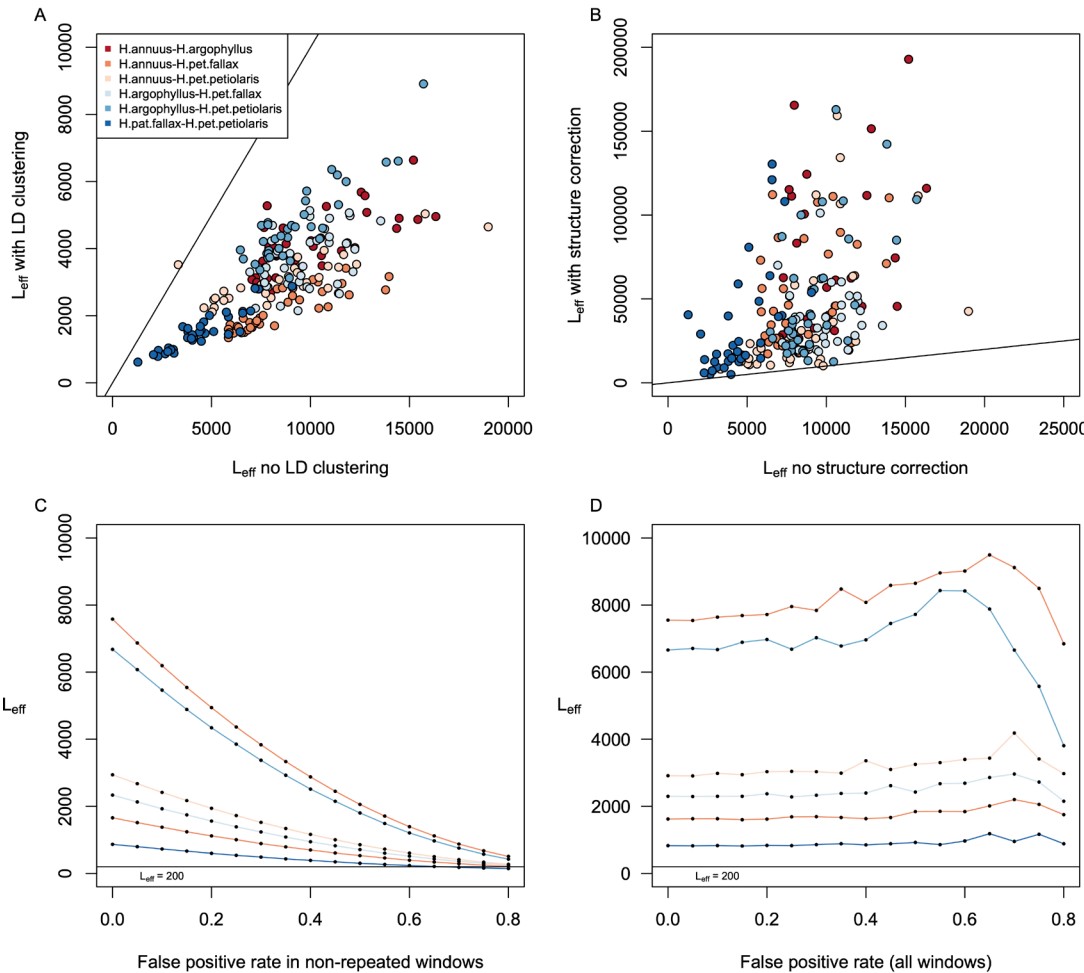

**Appendix 1—figure 15.** Estimates of effective number of loci in pairwise contrasts among species. Panel (**A**) shows a comparison of estimates of the effective number of loci ($L_{eff}$) when calculated with vs. without linkage disequilibrium (LD)-clustering for the environmental variables from the six pairwise contrasts among lineages. Panel (**B**) shows the effect of structure correction using BayPass on $L_{eff}$. Panels (**C**) and (**D**) show the estimation of $L_{eff}$ for the variable with the lowest average value (Hargreaves reference evapotranspiration; Eref) under different false-positive rates for just the windows with non-repeated signatures (**C**) or for all windows (**D**).

