## [Editor Report · eLife assessment]

This is a **valuable** comparative study of adaptation across multiple species. The results provide a **solid** example of the application of genotype–environment associations to demonstrate that local adaptation is repeatable.

---

## [Referee Report · Reviewer #1 (Public Review)]

Soudi, Jahani et al. provide a valuable comparative study of local adaptation in four species of sunflowers, and investigate the repeatability of observed genomic signals of adaptation and their link to haploblocks, known to be numerous and important in this system. The study builds on previous work in sunflowers that have investigated haploblocks in those species and on methodologies developed to look at repeated signals of local adaptations. The authors provide solid evidence of both genotype-environment associations (GEA) and genome-wide association study (GWAS), as well as phenotypic correlations with the environment, to show that part of the local adaptation signal is repeatable and significantly co-occur in regions harboring haploblocks. Results also show that part of the signal is species specific and points to high genetic redundancy. This work will be of interest to evolutionary biologists in general and population geneticists in particular, and constitutes a good example of comparative local adaptation. Importantly, this study helps in advancing our understanding of the genetic architecture implicated in the adaptation process.

Strenghts: The authors take great care in acknowledging and investigating the multiple biases inherent to the used methods (GEA and GWAS) and use conservative and well thought statistical approaches to draw their conclusions. Additionally, I appreciated the nuanced discussion and can only agree with the authors that the adaptation process is complex and does not fully fit the classic simplified genetics models of either few large effect genes or only infinitesimal quantitative traits. I find the added Summary figure of this revised version (S1) extremely helpful in better understanding the different analysis steps and how they relate to the different questions.

Weaknesses: After those revisions, I did not find any major weakness and am satisfied with the authors responses.

---

## [Referee Report · Reviewer #2 (Public Review)]

In this study the authors sought to understand the extent of similarity among species in intraspecific adaptation to environmental heterogeneity at the phenotypic and genetic levels. A particular focus was to evaluate if regions that were associated with adaptation within putative inversions in one species were also candidates for adaptation in another species that lacked those inversions. This study is timely for the field of evolutionary genomics, due to recent interest surrounding how inversions arise and become established in adaptation.

Major strengths-

Their study system was well suited to addressing the aims, given that the different species of sunflower all had GWAS data on the same phenotypes from common garden experiments as well as landscape genomic data, and orthologous SNPs could be identified. Organizing a dataset of this magnitude is no small feat. The authors integrate many state-of-the-art statistical methods that they have developed in previous research into a framework for correlating genomic Windows of Repeated Association (WRA, also amalgamated into Clusters of Repeated Association based on LD among windows) with Similarity In Phenotype-Environment Correlation (SIPEC). The WRA/CRA methods are very useful and the authors do an excellent job at outlining the rationale for these methods.

Weaknesses-

The authors did an excellent job responding to the first set of reviews and overall I found the manuscript more streamlined and easier to read. The main weakness in the manuscript is that correlations among environmental variables were not controlled for in their results, and is a source of potential pseudoreplication. The authors are clear about the results that are affected by pseudoreplication.

The manuscript shows how to integrate many recent methods to study the repeatability of adaptation, and the methods and data are likely to be used in similar studies.

---

## [Author Response]

The following is the authors’ response to the original reviews.

**Reviewer #1 (Public Review):**
Soudi, Jahani et al. provide a valuable comparative study of local adaptation in four species of sunflowers and investigate the repeatability of observed genomic signals of adaptation and their link to haploblocks, known to be numerous and important in this system. The study builds on previous work in sunflowers that have investigated haploblocks in those species and on methodologies developed to look at repeated signals of local adaptations. The authors provide solid evidence of both genotype-environment associations (GEA) and genome-wide association study (GWAS), as well as phenotypic correlations with the environment, to show that part of the local adaptation signal is repeatable and significantly co-occur in regions harboring haploblocks. Results also show that part of the signal is species specific and points to high genetic redundancy. The authors rightfully point out the complexities of the adaptation process and that the truth must lie somewhere between two extreme models of evolutionary genetics, i.e. a population genetics view of large effect loci and a quantitative genetics model. The authors take great care in acknowledging and investigating the multiple biases inherent to the used methods (GEA and GWAS) and use a conservative approach to draw their conclusions. The multiplicity of analyses and their interdependence make them slightly hard to understand and the manuscript would benefit from more careful explanations of concepts and logical links throughout. This work will be of interest to evolutionary biologists and population geneticists in particular, and constitutes an additional applied example to the comparative local adaptation literature.Some thoughts on the last paragraph of the discussion (L481-497): I think it would be fine to have some more thoughts here on the processes that could contribute to the presence/absence of inversions, maybe in an "Ideas and Speculation" subsection. To me, your results point to the fact that though inversions are often presented as important for local adaptation, they seem to be highly contingent on the context of adaptation in each species. First, repeatability results are only at the window/gene level in your results, the specific mutations are not under scrutiny. Is it possible that inversions are only necessary when sets of small effect mutations are used, opposite to a large effect mutation in other species? Additionally, in a model with epistasis, fitness effects of mutations are dependent on the genomic background and it is possible that inversions were necessary in only certain contexts, even for the same mutations, i.e. some adaptive path contingency. Finally, do you have specific demographic history knowledge in this system that maps to the observations of the presence of inversions or not? For example, have the species "using" inversions been subject to more gene flow compared to others?

Thank you for the great suggestions and helpful comments. Regarding the question of demography, each of the species actually harbours quite a large number of haploblocks (13 in H. annuus spanning 326Mb, 6 in H. argophyllus spanning 114 Mb, and 18 in H. petiolaris spanning 467 Mb; see Todesco et al. 2020 for more details) so there does not seem to be any clear association with demography. We agree about the complexities that might underly the evolution of inversions that you outline above, and have refined some of the text where we discuss their evolution in the Discussion.

**Reviewer #2 (Public Review):**
In this study the authors sought to understand the extent of similarity among species in intraspecific adaptation to environmental heterogeneity at the phenotypic and genetic levels. A particular focus was to evaluate if regions that were associated with adaptation within putative inversions in one species were also candidates for adaptation in another species that lacked those inversions. This study is timely for the field of evolutionary genomics, due to recent interest surrounding how inversions arise and become established in adaptation.Major strengthsTheir study system was well suited to addressing the aims, given that the different species of sunflower all had GWAS data on the same phenotypes from common garden experiments as well as landscape genomic data, and orthologous SNPs could be identified. Organizing a dataset of this magnitude is no small feat. The authors integrate many state-of-the-art statistical methods that they have developed in previous research into a framework for correlating genomic Windows of Repeated Association (WRA, also amalgamated into Clusters of Repeated Association based on LD among windows) with Similarity In Phenotype-Environment Correlation (SIPEC). The WRA/CRA methods are very useful and the authors do an excellent job at outlining the rationale for these methods.

Thank you!

Major weaknessesThe study results rely heavily on the SIPEC measure, but I found the values reported difficult to interpret biologically. For example, in Figure 4 there is a range of SIPEC from 0 to 0.03 for most species pairs, with some pairs only as high as ~0.01. This does not appear to be a high degree of similarity in phenotype-environment correlation. For example, given the equation on line 517 for a single phenotype, if one species has a phenotype-environment correlation of 1.0 and the other has a correlation of 0.02, I would postulate that these two species do not have similar evolutionary responses, but the equation would give a value of (1+0.02)*1*0.02/1 = 0.02 which is pretty typical "higher" value in Figure 4. I also question the logic behind using absolute values of the correlations for the SIPEC, because if a trait increases with an environment in one species but decreases with the environment in another species, I would not predict that the genetic basis of adaptation would be similar (as a side note, I would not question the logic behind using absolute correlations for associations with alleles, due to the arbitrary nature of signing alleles). I might be missing something here, so I look forward to reading the author's responses on these thoughts.

The reviewer makes a very good point about the range of SIPEC, and we have changed our analysis to reflect this, now reporting the maximum value of SIPEC for each environment (across the axes of the PCA on phenotypes that cumulatively explain 95% of the variance), in Figure 4 and Supplementary Figures S2 and S13. For consistency among manuscript versions and to illustrate the effect of this change, we retain the mean SIPEC value in one figure in the supplementary materials (S12), which shows the small effect of this change on the qualitative patterns. Figure 4 now shows that the maximum SIPEC value is regularly quite strong, which should address the reviewer’s concern that this is not being driven by anomalous and small values. We appreciate this point and think this change now more closely reflects how we are trying to estimate the biological feature of interest – that some axis of phenotypic space is strongly (or not) responding to selection from the environmental variable.

With respect to the logic behind using absolute value, we still feel this is justified for traits, because if a trait evolves to be bigger or smaller, it may still use the same genes. For example, flowering time may change to be later or earlier, which would result in opposite correlations with a given environment, but might use the same gene (e.g. FT) for this. As such, we think keeping absolute value is more representative as otherwise species with strong but opposite patterns of adaptation would look like they were very different. We have added a statement on line 584 in the methods section to further clarify the reason for this choice.

An additional potential problem with the analysis is that from the way the analysis is presented, it appears that the 33 environmental variables were essentially treated as independent data points (e.g. in Figure 4, Figure 5). It's not appropriate to treat the environmental variables independently because many of them are highly correlated. For example in Figure 4, many of the high similarity/CRA values tend to be categorized as temperature variables, which are likely to be highly correlated with each other. This seems like a type of pseudo replication and is a major weakness of the framework.

This is a good point and we fully agree. It is for this reason that we didn’t present any p-values or statistical tests of the overall patterns that are shown in these figures (i.e. the linear relationship between SIPEC and number of CRAs in figure 4 and the tendency for most points to fall above the 1:1 line in figure 5). But to make sure this is even more clear, we have added statements to the captions of these figures to remind readers that points are non-independent. We still feel that in the absence of a formal test, the overall patterns are strongly consistent with this interpretation. A smaller number of non-pseudo-replicated points in Figure 4 would still likely show linear patterns. Similarly, there are almost no significant points falling below the 1:1 line in Figure 5, and it seems unlikely that pseudoreplication would generate this pattern.

Below I highlight the main claims from the study and evaluate how well the results support the conclusions."We find evidence of significant genome-wide repeatability in signatures of association to phenotypes and environments" (abstract)Given the questions above about SIPEC, I did not find this conclusion well supported with the way the data are presented in the manuscript.

We have changed the reporting of the SIPEC metric so that it more clearly reflects whichever axis of phenotypic space is most strongly correlated with environment in both species (using max instead of mean). This shows similar qualitative patterns but illustrates that this happens across much higher values of SIPEC, showing that it is in fact driven by high correlations in each species (or non-similar correlations resulting in low values of SIPEC). While we agree about the pseudo-replication problem preventing formal statistical test of this hypothesis, the visual pattern is striking and seems unlikely to be an artefact, so we think this does still support this conclusion.

"We find evidence of significant genome-wide repeatability in signatures of association to phenotypes and environments, which are particularly enriched within regions of the genome harbouring an inversion in one species. " (Abstract) And "increased repeatability found in regions of the genome that harbour inversions" (Discussion)*WRAs*, not whether *haploblocks*. The wording of the abstract is claiming the latter, but I think what they tested was the former. Let me know if I'm missing something here.These claims are supported by the data shown in Figure 4, which shows that haploblocks are enriched for WRAs. I want to clarify a point about the wording here, as my understanding of the analysis is that the authors test if *haploblocks* are enriched with *WRAs* are enriched for

We are actually not interested in whether WRAs are enriched for haploblocks; we want to know if WRAs tend to occur more commonly within haploblocks than outside of them. We have tried to clarify that this is our aim in various places in the manuscript. Our analysis for Figure 5 is the one supporting these claims, and it uses the Chi-square test statistic to assess the number of WRAs and non-WRAs that fall within vs. outside of inversions, and a permutation test to assess the significance of this observation, for each environmental variable and phenotype. We don’t think that this test has any direction to it – it’s simply testing if there is non-random association between the levels of the two factors. Thus, we think the wording we have used is consistent with the test result and our aims. Perhaps the confusion arose from the two methods that we present in the Methods (one is used for Figure 5, the other for Figure S6C & D), so we have added clarifications there.

Notwithstanding the concerns about highly correlated environments potentially inflating some of the patterns in the manuscript, to my knowledge this is the first attempt in the literature to try this kind of comparison, and the results does generally suggest that inversions are more likely capturing, rather than accumulating adaptive variation. However, I don't think the authors can claim that repeated signatures are enriched with haploblock regions, and the authors should take care to refrain from stating the relative importance of different regions of the genome to adaptation without an analysis.

Actually, we don’t have a strong feeling about whether inversions are capturing vs. accumulating adaptive variation, as these results could be consistent with either. As described above, we do not understand why we can’t claim that repeated signatures are enriched within haploblocks. We thought the reviewer is perhaps referring to the fact that the points are pseudo-replicated in the figures due to environment? We note that a very large number of points are significantly different from random in terms of the distribution of WRAs within vs. outside of haploblocks (light- vs. dark-shaded symbols), and that almost all of them fall above the 1:1 line. While there may be pseudo-replication preventing a test of the bigger multi-environment/multi-species hypothesis across all phenotypes and environments, there is almost a complete lack of significant results in the other direction. This seems like quite strong evidence about enrichment of WRAs within haploblocks, across many environments/species contrasts. We have added some text to the description of patterns in figure 5 to try to clarify this.

"While a large number of genomic regions show evidence of repeated adaptation, most of the strongest signatures of association still tend to be species-specific, indicating substantial genotypic redundancy for local adaptation in these species." (Abstract)Figure 3B certainly makes it look like there is very little similarity among species in the genetic basis of adaptation, which leaves the question as to how important the repeated signatures really are for adaptation if there are very few of them. (Is 3B for the whole genome or only that region?). This result seems to be at odds with the large number of CRAs and the claims about the importance of haploblock regions to adaptation, which extend from my previous point.

Figure 3B is for the whole genome, we have added text to the figure caption to clarify this. We think that both interpretations are possible: that most of the regions of the genome that are driving adaptation are non-repeated, but that a small but significant proportion of regions driving adaptation are repeated above what would be expected at random. Thus, it seems that there is high redundancy, coupled with adaptation via some genes that seem particularly functionally important and non-redundant, and therefore repeated. We added clarifying text on lines 541-548.

"we have shown evidence of significant repeatability in the basis of local adaptation (Figure 4,5), but also an abundance of species-specific, non-repeated signatures (Figure 3)"While the claim is a solid one, I am left wondering how much of these genomes show repeated vs. non-repeated signatures, how much of these genomes have haploblocks, and how much overlap there really is. Finding a way to intuitively represent these unknowns would greatly strengthen the manuscript.

We agree, and really struggled to find the best way to communicate both the repeated patterns and the large amount of non-repeated signatures. Unfortunately, we have more confidence in the validity of repeated patterns because for the non-repeated patterns, a strong signature of association to environment in only one species could just be the product of structureenvironment correlation, as we didn’t control for population structure. Thus, trying to quantify the proportion of non-repeated signatures is difficult to do with any accuracy and we preferred to avoid putting too much emphasis on the simple calculation of the proportion of top candidate windows that were also WRAs.

Overall, I think the main claims from the study, the statistical framework, and the results could be revised to better support each other.Although the current version of the manuscript has some potential shortcomings with regards to the statistical approaches, and the impact of this paper in its present form could be stifled because the biology tended to get lost in the statistics, these shortcomings may be addressed by the authors.With some revisions, the framework and data could have a high impact and be of high utility to the community.

Thank you for your very helpful comments and suggestions on our paper, we really appreciate it.

Recommendations for the authors: please note that you control which revisions to undertake from the public reviews and recommendations for the authorsEditor's comments:The reviewers make a series of reasonable suggestions that I echo. I found the paper quite hard to follow, and got fairly lost in the various layers of analyses done. Partially, this represents the complexity of empirical genomic data, which rarely deliver simple stories of convergence at a few genes. However, the properties of the various statistics used to detail local adaptation and convergence are not particularly clear and the figures presented were not intuitive representations of the data. This leaves the reader with an incomplete view of how much weight to put in the various lines of evidence marshaled. I would suggest simplifying the presentation of the results considerably. I add a few additional comments below.

Great suggestion, we’ve added a schematic overview of the methods and main research questions to Figure S1 in the supplementary materials.

A figure would help showing some of the signals of SNPs with putative signals of convergent environmental correlations across species, e.g. frequencies plotted against climate variables. This would help readers get a sense of how strong these signals were. These could be accompanied by the statistics calculated for these SNPs, that would allow the reader to start to get some intuitive sense of what the numbers mean.

Great suggestion, we have added a schematic overview of the methods to Figure S1 that shows some of the values and illustrates how the methods work using visual examples from our data.

In general, the introduction and some of the discussion of the inversion results feel oddly framed:Abstract line 36: "This shows that while inversions may facilitate local adaptation, at least some of the loci involved can still make substantial contributions without the benefit of recombination suppression."

We have changed “some of the loci involved can still make substantial contributions without the benefit of recombination suppression” here to “some of the loci involved can still harbour mutations that make substantial contributions without the benefit of recombination suppression in species lacking a segregating inversion” as it hopefully clarifies that we’re not talking about individual alleles that are present in both species.

Models of the role of local adaptation in the establishment of inversions (Kirkpatrick & Barton) assume that there are multiple locally adapted alleles already present. It is the load created by these alleles being constantly maintained in the face of migration and subsequent recombination that allow an inversion to be selected for because it keeps together locally adapted alleles. Thus these models predict that there could well be standing local adaptation at these loci in the absence of the inversion in other species, and that these locally adapted alleles while not fixed may be at high frequency. (After establishment, inversions housing locally adapted alleles, can shield more weakly, locally beneficial alleles from migration allow other alleles to build up.) Empirically it's interesting to find signals of local adaptation in other species that don't contain putative inversions. But the logic of the different predictions is not particularly clear from the introduction, and only becomes somewhat clearer in the discussion.

Thank you for pointing out this murkiness, we have re-written portions of both the Introduction and Discussion to clarify this aspect.

From the introduction:Inversions have been implicated in local adaptation in many species (Wellenreuther and Bernatchez 2018), likely due to their effect to suppress recombination among inverted and noninverted haplotypes, and thereby maintain LD among beneficial combinations of locally adapted alleles (Rieseberg 2001; Noor et al. 2001; Kirkpatrick and Barton 2006). This has been approached by models studying the establishment of inversions that capture combinations of locally adapted alleles present as standing variation (e.g., Kirkpatrick and Barton 2006), as well as models examining the accumulation of locally adapted mutations within inversions (e.g., Schaal et al. 2022). If there is variation in the density of loci that can potentially contribute to local adaptation, inversions would be expected to preferentially establish and be retained in regions harbouring a high density of such loci (and this expectation would hold for both the capture and accumulation models). We would also expect to see stronger signatures of repeated local adaptation in such high density regions. Despite mounting evidence of their importance in adaptation, it is unclear how inversions may covary with repeatability of adaptation among species. A fundamental parameter of importance in these models is the relationship between migration rate and strength of selection on individual alleles, which may not make persistent contributions to local adaptation without the suppressing effects of recombination if selection is too weak (Yeaman and Whitlock 2011; Bürger and Akerman 2011). If most alleles have small effects relative to migration rate and can only contribute to local adaptation via the benefit of the recombination-suppressing effect of an inversion, then we would expect little repeatability at the site of an inversion – other species lacking the inversion would not tend to use that same region for adaptation because selection would be too weak for alleles to persist. On the other hand, if some loci are particularly important for local adaptation and regularly yield mutations of large effect, with these patterns being conserved among species, repeatability within regions harbouring inversions may be substantial. Thus, studying whether adaptation at the same genomic region harbouring an inversion is observed in other species lacking the inversion can give insights about the underlying architecture of adaptation, and the evolution and maintenance of inversions.

From the Discussion:The observed repeatability associated with inversions further supports the local adaptation model as an explanation for the long-term persistence of segregating inversions at least in sunflowers, rather than mechanisms based on dominance or meiotic drive (Rieseberg 2001). If there is variation across the genome in the density of loci with the potential to be involved in local adaptation, then the establishment and maintenance of inversions would be biased towards regions harbouring a high density such loci under this model. If the genomic basis for local adaptation is conserved amongst species, then these same regions are more likely to have high repeatability. Thus, our observation of genomic regions harbouring inversions also being enriched for WRAs is consistent with this general model for inversion evolution. Unfortunately, our observations do not provide much insight into whether inversions evolve through the capture (e.g. Kirkpatrick and Barton 2006) or accumulation (e.g. Schaal et al. 2022) type of model, as either model would be consistent with our results. Most of the sunflower inversions are >1 My old, and therefore predate any current local adaptation patterns, but likely do not predate the genes underlying local adaptation (which appear to be shared among the species we studied). As for the alleles underlying local adaptation, they may be younger than the inversions, but as our work suggests, these regions are prone to harbouring locally adaptive alleles so it is possible that they also harboured other ancestral locally adaptive alleles.

As a minor comment, there's a fair number of places where a more nuanced view of the field is needed, e.g.:"Models in evolutionary genetics tend to focus on extremes: population genetic approaches explore cases where strong selection deterministically drives a change in allele frequency" --This seems like a strange strawman. Population genetic models span a huge parameter range. The empirical approaches of looking for sweeps by detecting genome-wide statistical outliers is predicated on strong selection, but there are numerous papers that have looked for signals of weak selection genome-wide.

Good point, we have changed our wording here.

**Reviewer #1 (Recommendations For The Authors):**
CommentsMy main comment on the manuscript is that the different levels and diversity of analyses are slightly hard to follow on the first, and even second, read. As there are several layers of correlations and comparisons, as well as some independent analyses, I wonder if it might be helpful to have a summary schematic figure of how all analyses fit together.

Great idea, we have added Figure S1 that summarizes the main flow of the methods and research questions.

L169-171: Would it be more accurate to say that SIPEC is maximized when both species have strong correlations for an environmental variable across the same phenotypes? But maybe I misunderstood the index.

Good point, we have now simplified SIPEC, reporting the max instead of the mean, which we think better reflects when similar patterns are happening in both species for some phenotype.

L191: Given the discussion in the introduction and elsewhere about the correction for population structure, which version is used here? Same for Figure 3.

We have added clarification there.

L348: One [environmental] variable?

Added

L353: Maybe add a percentage indication for 387 so that it is comparable to the following23.3%.

Good point, added

-> L388 and paragraph: You mention "significant repeatability" but it is hard from the results at this point to have a broad idea of the amount of signal that is repeatable. Would it be possible to add here some quantitative measure of the proportion of signal repeatable or not, even if approximated?

I wish we could, but I think the precision implied by such an approximation would involve a huge amount of uncertainty and likely inaccuracy. Because it is so hard to conclusively identify how many loci are significant but non-repeated, we really don’t have a good handle on the denominator here. We are pretty confident that the repeated loci are strongly enriched for true positives, but the non-repeated loci are also almost certainly strongly enriched for false positives. While we really want to be able to quantify this explicitly, we don’t think it’s possible given our data.

-L415-418: "If there is variation [...] involved in local adaptation", I do not follow this argument, could you rephrase?

Changed

-L447-450: As you say in the supplementary methods, your analyses exclude 3/4 of the genome.Do you think this choice has a large impact on the number of outliers observed here as the genome-wide baseline would change?

This is a very good question, but one that is quite complex and without a clear answer – we chose not to delve into it in the paper to keep the discussion streamlined. My (SY) feeling is that it is unlikely that regions harbouring transposable elements would contribute much to adaptation, but I think we really don’t know if that is true. Even excluding ¾ of the genome harbouring TEs, ¼ of the genome still constitutes a huge amount of sequence and a very large number of genes and it seems plausible that most genes and genic regions would not contribute to adaptation for a given trait, so I don’t think this would change the results too much in a qualitative way – but would almost certainly change the number of windows that are significant, etc.

L455-457: "As we are unable [...] potentially important drivers" Could you provide the logical link here between loci of small effect and them being important drivers. I presume you mean that the large effect loci found here only account for a small proportion of the heritability?

Yes that’s what we meant here, so we’ve added some clarification.

L482: "enriched within inversions" should that be 'in genomic regions where there exist inversions in at least one species'? Thanks for catching that, yes. Changed.Methods/SIPEC L512: Compared to the Results section it is unclear here what is referred to as an "environment" Is it a variable or a set of environment variables?

This is done per environmental variable.

I find the presence of the PCA for environment variables in Figure 2 misleading as my first interpretation was that PCs for environment were also used.

Good point, we have clarified this on line 190-193.

Maybe one potential addition to the formula would be to add an environment variable $j$ notation such that it reads "$SIPEC_j = \sum_i (|r_{ij,1}| + ...) ...$ where ... between environment variable $j$". I had initial difficulties to understand how this SIPEC was computed relating to environmental variables and this might help.

Given the other changes we made to SIPEC, we felt it was simpler to just present it as a single calculation on a given combination of phenotype and environment for a pair of species, and then discuss taking the mean and maximum of this later.

Finally, PCA axes explaining 95% of the variance are used, I would find it interesting to see how many PCs are used in comparison to the number of traits being measured.

We have added the following sentence to the methods describing this:

"For comparisons including H. argophyllus, 95% of the variance was typically explained by 8-10 PC axes (out of 28 or 29 phenotypes), whereas for comparisons among other taxa this included 21 or 22 PC axes out of 65 or 66 phenotypes."

TyposL52: --

Changed

L254: portions [of] their

Changed

L399: additional closing parenthesis

Changed

L458: signatures [of] repeated association

Changed

L554: performed [on]

Changed

L578: 5 kp/kb windows

Changed

L601: casual/causal SNPs

Changed

L615: widow/window

Changed

L732: Banding/Banting Postdoctoral Fellowship

Changed

L1002 & L960: [Supplementary] Figure

Changed

Supplementary: Some figure titles are in bold and others are not.

Changed

**Reviewer #2 (Recommendations For The Authors):**
Overall I found the writing to be very clear and easy to follow. Despite my comments, it was clear that a lot of thought went into how to conduct the tests and visualize the results. I recommend ending the Discussion on a positive note, rather than an impossible test.

Thanks for the positive suggestion, we have done this.

In Figure 5, is the temperature variable missing in the legend and in the plot?

No, for this plot we just combined the temperature/precipitation variables into one variable called “climate”.